



# Glacier runoff variations since 1955 in the Maipo River Basin, semiarid Andes of central Chile

Álvaro Ayala[1,2], David Farías-Barahona[3], Matthias Huss[1,2,4], Francesca Pellicciotti[2,5], James McPhee[6,7], Daniel Farinotti[1,2]

[1]Laboratory of Hydraulics, Hydrology and Glaciology (VAW), ETH Zurich, Zurich, 8093, Switzerland.
[2]Swiss Federal Institute for Forest, Snow and Landscape Research, Birmensdorf, 8903, Switzerland.
[3]Institut für Geographie, Friedrich-Alexander-Universität Erlangen-Nürnberg, Erlangen, 91058, Germany
[4]Department of Geosciences, University of Fribourg, Fribourg, 1700, Switzerland
[5]Department of Geography, Northumbria University, Newcastle, NE1 8ST, UK.
[6]Department of Civil Engineering, University of Chile, Santiago, 8370449, Chile.
[7]Advanced Mining Technology Centre (AMTC), University of Chile, Santiago, 8370451, Chile.

*Correspondence to*: Alvaro Ayala (alvaro.ayala@ceaza.cl), now at Centre for Advanced Studies in Arid Zones (CEAZA)

**Abstract (max 250 words).** As glaciers adjust their size in response to climate variations, long-term changes in meltwater production can be expected, affecting the local availability of water resources. We investigate glacier runoff in the period

1955-2016 in the Maipo River Basin (4 843 km$^2$), semiarid Andes of Chile. The basin contains more than 800 glaciers covering 378 km$^2$ (inventoried in 2000). We model the mass balance and runoff contribution of 26 glaciers with the physically-oriented and fully-distributed TOPKAPI-ETH glacio-hydrological model, and extrapolate the results to the entire basin. TOPKAPI-ETH is run using several glaciological and meteorological datasets, and its results are evaluated against streamflow records, remotely-sensed snow cover and geodetic mass balances for the periods 1955-2000 and 2000-2013. Results show that glacier

mass balance had a general decreasing trend as a basin average, but with differences between the main sub-catchments. Glacier volume decreased by one fifth (from 18.6±4.5 to 14.9±2.9 km$^3$). Runoff from the initially glacierized areas was 186±27 mm yr$^{-1}$ (17±7% of the total contributions to the basin), but it shows a decreasing sequence of maxima, which can be linked to the interplay between a decrease in precipitation since the 1980s and the reduction of ice melt. If glaciers in the basin were in equilibrium with the climate of the last two decades, their volume would be reduced to 81±38% of the year 2000 volume, and

glacier runoff during dry periods would be 61±24% of its maximum contribution in the period 1955-2016, considerably decreasing the drought mitigation capacity of the basin.



## 1 Introduction

Most glaciers on Earth have retreated due to global atmospheric warming during the 20[th] century (Zemp et al., 2019). Glaciers
that are still out of balance with the present climate are committed to lose part of their mass in the coming decades, even
without further warming (Zemp et al., 2015; Marzeion et al., 2018), and major changes in their meltwater production can be
anticipated (Bliss et al., 2014; Huss and Hock, 2018; IPCC, 2019). In the absence of precipitation changes, a temporary increase
of meltwater generation from a retreating glacier occurs as a consequence of higher air temperatures and enhanced ablation,
but after this transient phase, melt amounts decrease due to the reduction of the available snow, firn and ice volumes (Jansson
et al., 2003). The period in which the annual melt volume reaches its long-term maximum has been termed "peak water"
(Gleick and Palaniappan, 2010; Baraer et al., 2012). Global-scale studies indicate a large heterogeneity in the geographical
distribution of peak water; while several catchments in Himalaya and Alaska are expected to increase their glacier runoff due
to the enhanced ablation in the next decades and reach a maximum at some point of the 21[st] century, other regions in the world,
such as the semiarid Andes, Central Europe and Western Canada, have already reached a regional maximum, and thus glacier
runoff will only decrease in the future (Bliss et al., 2014; Huss and Hock, 2018). While these studies provide global trends that
are key for macro-regional assessments, studies focusing on the catchment-scale can provide more specific information about
local hydro-glaciological changes to communities and stakeholders for the generation of mitigation and adaptation strategies.
Additionally, catchment-scale studies place glacier runoff in the context of other components of the water cycle, and evaluate
the impacts of glacier changes on downstream areas.

In this study, we focus on glacier changes and their impacts on long-term glacier runoff contribution in the semiarid Andes.
Melt water originated in the Andes are key for Chile and the western areas of Argentina, as it represents the main source for
drinking water, agriculture, industry, mining and ecosystems. The climate of this region is characterized by its strong inter-
annual variability of precipitation linked to periodic atmosphere-ocean variations over the Pacific Ocean (Montecinos and
Aceituno, 2003; Falvey and Garreaud, 2007) and a sustained air temperature increase during the last decades (Carrasco et al.,
2005; Burger et al., 2018b). A few studies have estimated the present (Ragettli and Pellicciotti, 2012; Ayala et al., 2016; Burger
et al., 2018a) and future (Ragettli et al., 2016; Huss and Hock, 2018) glacier runoff contribution in the semiarid Andes, but its
past variations have not been analysed in detail, mostly due to the lack of long-term glaciological data. As future climate
scenarios anticipate a decrease in glacier runoff (e.g. Ragettli et al., 2016), the question of whether peak water has already
occurred still remains open. The assessment of long-term changes in glacier runoff is particularly useful for water planners,
because it provides reference information for the role of glacier meltwater in river flows, and the impacts that can be anticipated
in the absence of its contribution.

Glaciers in the semiarid Andes underwent a major retreat in the 20[th] century (Le Quesne et al., 2009; Malmros et al., 2016),
and the last two decades (Braun et al., 2019; Dussaillant et al., 2019). Historical documents, aerial photographs, and
dendrochronological studies suggest that the general retreat trend started around the mid-19[th] century, but it has been





interrupted by occasional periods of positive mass balance accompanied by glacier advances (Le Quesne et al., 2009; Masiokas et al., 2009). Masiokas et al. (2016) performed a reconstruction of the annual mass balance of Echaurren Norte Glacier (3650-3900 m a.s.l.) since 1909 using a simple glacier mass-balance model forced with monthly precipitation and air temperature. The model was verified against streamflow records and direct mass balance measurements on the glacier, where the first mass balance monitoring programme in the Southern Andes started in 1975. Masiokas et al. (2016) found a general retreat

interrupted by three periods of sustained positive mass balances in the 1920-30s, 1980s and 2000s. The latter, positive or balanced mass budget in the semiarid Andes in the 2000-2009 period has been recently verified by geodetic mass balances (Braun et al., 2019; Dussaillant et al., 2019; Farías-Barahona et al., 2019b), and has been supported by independent modelling results (Burger et al., 2018a). As the findings by Masiokas et al. (2016) are based on a relatively simple model applied to only one glacier at low elevation (<4000 m a.s.l.), they cannot be extrapolated to other glaciers. This is especially true due to the

large spatial variability of response times and retreat rates reported for this region (Malmros et al., 2016). Thus, a more detailed analysis based on the specific characteristics of each glacier is needed to complement these results and estimate regional changes of ice volume and glacier runoff. From a climatic perspective, glacier retreat in the semiarid Andes has been driven by a general temperature increase and modulated by a strong temporal variability of precipitation. Air temperature showed an increasing trend of about 0.25°C decade$^{-1}$ in the period 1979-2006 (Falvey and Garreaud, 2009), mostly explained by a spring

and autumn warming (Burger et al., 2018b), which can be used to explain an increase of the 0°C isotherm and the regional Equilibrium Line Altitude (ELA) (Carrasco et al., 2005, 2008). Precipitation, on the other hand, exhibited an average decrease of –65 mm (–7.1%) decade$^{-1}$ in the period 1979-2016 (Boisier et al., 2016), although with a large inter-annual and -decadal variability (Montecinos and Aceituno, 2003).

   Our main objective is to reconstruct glacier changes during the last six decades in one of the main catchments of the semiarid

Andes, the Maipo River Basin, analyse the role of glaciers in the regional hydrology, and identify the main trends in glacier runoff. Additionally, we estimate glacier changes under synthetic scenarios of committed ice loss, in which air temperature, precipitation and cloudiness are assumed to stay at their current levels until the end of the century. Such a scenario can be used to compute the minimum changes that glaciers will experience due to past changes of the climate. They are thus highly conservative and do not correspond to a realistic projection for the future. The calculation of glacier changes and runoff

contribution is carried out for a subset of the largest glaciers using the physically-oriented and fully-distributed TOPKAPI-ETH glacio-hydrological model (Ayala et al., 2016; Ragettli et al., 2016), and the resulting mass balances are extrapolated to the entire basin (Huss, 2012). We set up the glacier model using glacier inventories, Digital Elevation Models (DEMs) and estimates of ice thickness, and we force it with a combination of local meteorological stations and reanalysis data for precipitation, air temperature and solar radiation. The model is calibrated and validated using remotely-sensed snow cover,

streamflow records, and geodetic mass balances covering the periods 1955-2000 and 2000-2013 (Braun et al., 2019; Farías-Barahona et al., 2019a).





## 2 Study area

The study focuses on the headwaters of the Maipo River Basin (for simplicity we hereafter refer to these areas as the Maipo River Basin). The basin is located in central Chile (~33°S, ~70°W), to the east of the Chilean capital city, Santiago (Figure
1a), to which it provides most of its drinking water. The basin outlet is the Maipo en El Manzano gauging station, which roughly marks the boundary between rural mountain areas and Santiago urban districts. The selected basin has an area of 4 843 km$^2$, its elevation ranges from 850 to 6570 m above sea level (a.s.l.), and more than 800 glaciers covering about 378 km$^2$ (7.8% glacierized) were inventoried in 2000 (Barcaza et al., 2017). The Maipo River and its tributaries are the primary source for drinking water, agriculture, hydropower, and industry in the region, which concentrates about 40% of the country's population.
The region has a Mediterranean-type climate, with a strong seasonality characterized by cold and wet winters, and hot and dry summers. Average precipitation in Santiago was 308 mm yr$^{-1}$ in the period 1950-2018, but values as low as 69 mm yr$^{-1}$ and as high as 712 mm yr$^{-1}$ have been registered, with a coefficient of variation of 0.45. Recurrent droughts have been reported since the beginning of hydro-meteorological records. Precipitation amounts are, in general, larger towards the south and towards higher elevations. An early study estimated that glacier runoff in the Maipo River Basin represents about 34% of the total
discharge in February, and up to 67% during summer months of dry years, such as 1968-1969 (Peña and Nazarala, 1987).

There are five major sub-catchments in the study area, from north to south: Olivares, Colorado, Yeso, Volcán and Upper Maipo (Figure 1b). According to the national Chilean inventory (Barcaza et al., 2017) (described in next section), the highest glacierized sites are in the Olivares and Colorado sub-catchments, with mean elevations between 4200 and 4500 m a.s.l., and some glaciers reaching elevations higher than 5500 m a.s.l. (Figure 1c). The Upper Maipo sub-catchment, on the other hand,
has the lowest-lying glaciers, with mean elevation varying between 3500 and 4000 m a.s.l., and several glaciers reaching elevations below 3000 m a.s.l. (Figure 1c). Glacierized areas vary from 40 km$^2$ in the Volcán sub-catchment to 99 km$^2$ in Colorado. Upper Maipo has the largest number of individual glaciers (348), and most of them correspond to low-elevation, rock and debris-covered glaciers (Figure 1d). In general, glacier size tends to decrease towards the south, with the largest glaciers being located in Olivares (Juncal Sur, Olivares Gama, and Olivares Beta glaciers), and on the slopes of Tupungatito
and Marmolejo volcanoes (Volcán Tupungatito, Azufre and Marmolejo glaciers) in the Colorado sub-catchment. Another series of relatively large glaciers corresponds to debris-covered ones, such as Pirámide, Loma Larga, and Cerro Castillo glaciers.

## 3 Data

### 3.1 Geographic and topographic information

Glacier outlines are extracted from the national Chilean inventory (Barcaza et al., 2017) and the Marangunic inventory (Marangunic, 1979). While the information for the Maipo River Basin in the national inventory was produced using two



satellite images from the Landsat Enhanced Thematic Mapper ETM+ of 2003, the Marangunic inventory was mostly based on aerial photographs taken in 1955 during a national geodetic programme, and maps presented by Lliboutry (1956) for the few missing areas. For consistency with the DEM obtained from the Shuttle Radar Topography Mission (SRTM) for the year 2000,

we assume that the outlines derived from the national inventory are also valid for that year. Additionally, the glacierized areas in the national inventory that are not identified as such in 1955 (mostly rock glaciers and debris-covered areas) are added to the 1955 inventory. In this study, we assign an error of 5% to the year 2000 inventory (Paul et al., 2013), and we double this value for the year 1955 inventory. Based on the resulting inventories, we estimate that the total glacier area changed from 558 km$^2$ in 1955 to 378 km$^2$ in 2000 (–32.3%), and that the number of individual glaciers decreased from 861 to 854. Although

some small glaciers might have effectively disappeared, the decreasing trend in the total number of glaciers is also balanced by the fragmentation of large glaciers, such as the Olivares Alfa glacier complex, into several smaller units (Malmros et al., 2016).

In addition to the glacier inventory, we generate a mask of debris-covered glacier areas from the same Landsat images that were used to produce the Chilean glacier inventory. For this, we use the semi-automatic method based on band ratio

segmentation of TM4 and TM5 Landsat bands (Paul et al., 2004), and we manually correct the results using Google Earth imagery. For the year 1955, we maintain the same debris-cover maps as in the 2000-2010 period, i.e. assuming that no major changes have occurred in the extension of debris cover, but we delete small debris-covered areas on the upper glacier areas.

In our analyses, we use the DEMs for years 1955, 2000 and 2013, and the geodetic mass balance datasets for the periods 1955-2000 and 2000-2013, calculated by Farías-Barahona et al. (2019a) and Braun et al. (2019). While the DEMs for 2000 and 2013

correspond to a part of the products generated in the study of Braun et al. (2019) for the entire South-American Andes, the DEM for 1955 and the geodetic mass balance for the period 1955-2000 was produced in the study of Farías-Barahona et al. (2019a), who extended the period of analysis of Braun et al. (2019) for the Maipo River Basin. Here, we provide a brief description of the derivation of these datasets, but more details are included in the supplementary information. The 1955 DEM was calculated from digitized 50-m contour lines of the 1:50'000 official Chilean cartography product, which was also obtained

from the 1955 geodetic programme. While the DEMs for the year 2000 were extracted from the SRTM product, the DEM of Maipo River Basin for 2013 was derived from TanDEM-X post-processed products (which for this region correspond to the year 2013). The DEMs were co-registered following Nuth and Kääb (2011). Errors from the geodetic mass balances were assessed over stable ground, and calculated using a standard error propagation procedure, including typical error sources such as radar penetration signal. Two glaciers (San Francisco and Mirador del Morado) were discarded from the geodetic mass

balance, because the original SRTM product was not available for those areas (only the void-filled product). As rock glaciers exhibit changes that are smaller than the estimated uncertainties, they were also discarded from the geodetic mass balance.



### 3.2 Ice thickness

Distributed glacier ice thickness in 2000 is estimated for all individual glaciers using the method of Huss and Farinotti (2012) with the glacier outlines and the SRTM DEM. Standard model parameters are used except for glaciers classified as debris-

covered or rock glaciers. For these two types of ice bodies, the parameter prescribing ice flux is substantially reduced to obtain thicknesses comparable to the direct thickness observations on the debris-covered Pirámide Glacier and its neighbouring rock glaciers (DGA, 2012). The obtained ice thickness estimates compare well with Ground Penetrating Radar (GPR) measurements (DGA, 2014) on Volcán Tupungatito (1685 data points) and Marmolejo (1544 data points) glaciers extracted from the Glacier Thickness Database (GlaThiDa) (Gärtner-Roer et al., 2014), for which we find a Root Mean Square Error (RMSE) of 9.8 and

8.5 m, respectively.

Once the distributed ice thickness is calculated for every glacier for the year 2000, we use the geodetic mass balance in the period 1955-2000 to estimate the ice thickness distribution in 1955. In this procedure, we find the problem that for some grid cells showing a positive elevation change from the geodetic mass balance for the 1955-2000 period, the ice thickness in year 2000 is too small, resulting in an inferred negative thickness. To avoid this, and obtain a meaningful 1955 ice thicknesses that

are consistent with both the geodetic mass balance and the glacier inventory, we assign the year 2000 ice thickness to 1955 in these grid cells and add the estimated positive elevation change. In this way, we obtain a corrected ice thickness value in 2000 for 4.8% of the glacierized area. As no geodetic mass balance was calculated for rock glaciers they are assumed to have the same thickness in 1955 as in year 2000. A similar result was found in the study of Bodin et al. (2010) for rock glaciers near Santiago. Finally, to calculate the 1955 ice thickness of small glaciers that are not included in the year 2000 inventory, we use

the 1955 glacier areas from the glacier inventory and a scaling relation to calculate mean ice thickness ($\bar{h}$) as a function of the glacier area ($S$), and assume average thickness to be valid for every grid cell in these glaciers:

$$\bar{h} = c \cdot S^{\gamma-1}, \tag{1}$$

where $\gamma$=1.357 and $c$=28.5 are standard parameters in the area-volume scaling theory (Chen and Ohmura, 1990).

At the basin-scale, we find a total ice volume of 18.6±4.1 km³ and 16.1±2.4 km³ for 1955 and 2000 (a change of −13.8%),

respectively. Based on the geodetic mass balances for period 2000-2013, we estimate a total ice volume of 15.2±3.2 km³ for year 2013 (a change of −18.4% relative to 1955). For the total ice volume of the investigated basin, we assume an uncertainty of 15%. This is between the values estimated by Huss and Farinotti (2012) for regional totals (~12%), and the value estimated by Farinotti et al. (2017) for individual glaciers (~21%).

### 3.3 Hydro-meteorological data

Precipitation and temperature data for the period 1979-2016 are derived from daily gridded products developed by the Centre for Climate and Resilience Research in Santiago, Chile (CR2, www.cr2.cl). These products were generated for a national water



balance study led by the Chilean directorate of water resources (DGA) (DGA, 2017; Álvarez-Garretón et al., 2018). The CR2 daily precipitation product was generated by means of a statistical downscaling of precipitation and moisture fluxes from the ERA-Interim reanalysis. The downscaling procedure is based on multiple linear regressions with topographic parameters,

which were calibrated with quality-controlled precipitation records. The CR2 temperature product was obtained using near-surface temperature from ERA-Interim and land surface temperature (LST) from the Moderate Resolution Imaging Spectroradiometer (MODIS), by means of multiple regression models using LST as the explanatory variable and validated with local observations. For our study, while the CR2 precipitation product is linearly interpolated from its original resolution (0.05°) to the spatial resolutions of our glacio-hydrological models (1 km and 100 m, see sections 4.1.2 and 4.1.3) to generate

monthly average maps, the CR2 temperature product is used to generate basin-scale daily temperature lapse rates.

Daily cloud transmissivity of solar radiation is calculated from the Chilean solar radiation database (http://www.minenergia.cl/exploradorsolar/) for the years 2004-2016 at the location of Embalse El Yeso meteorological station, which is placed close to the centroid of the Maipo River Basin, and assumed to be uniform over the catchment. The solar radiation database was derived using reanalysis data to force a radiative transfer model for clear-sky solar irradiance and

an empirical model based on satellite data for cloudy conditions (Molina et al., 2017).

In addition to the information from the CR2 products, we use local records of air temperature and precipitation from Embalse El Yeso and Quinta Normal (located in Santiago) meteorological stations, respectively, as a base for extrapolating these variables during the period 1955-1978 (see sections 4.1.2 and 4.1.3). Values for air temperature gradients and cloud transmissivity in the study periods without information from CR2 and the Chilean solar radiation database (1955 to 1978 and

1955 to 2003, respectively) are randomly selected from a pool of values recorded in the same day of the year in the periods with available information. Finally, streamflow data for the Maipo River Basin are available as monthly mean records at the gauging station of Maipo en El Manzano. These time series were already corrected for extractions and reservoirs to approximate the natural flow in the study of CONIC-BF (2008).

### 3.4 Additional datasets

To calibrate and validate the snow processes in the study area, we use two products: (i) post-processed MODIS snow-cover area (SCA), downloaded from an online platform (http://www.dgf.uchile.cl/rene/MODIS/) that automatically calculates SCA from MODIS Terra and Aqua satellites in several Chilean river basins, and (ii) daily basin-scale snow water equivalent (SWE) estimates for the period 1984-2014, extracted from the Chilean version of the Catchment Attributes and Meteorology for Large Sample Studies (CAMELS-CL) database (http://camels.cr2.cl/). These basin-scale SWE estimates were aggregated by

Álvarez-Garretón et al. (2018) from a daily gridded product for the Andes Cordillera generated by Cortés et al. (2016) at a 180-m resolution using a data assimilation framework of the Modern Era Retrospective Analysis for Research and Applications (MERRA) reanalysis and Landsat imagery.





For modelling evapotranspiration and sub-surface water fluxes, we generate land use and soil types maps, respectively. The land use maps are extracted from the National Forest Corporation (CONAF) database (CONAF, 2013), and the same maps are

used to estimate the spatial distribution of soil types in the basin. For simplicity, and due to the absence of more detailed data, we define only two soil types based on the presence or absence of vegetation. The vegetated soil type dominates areas at low elevations and close to streams, whereas the no-vegetated one dominates on mountain slopes.

## 4 Methods

### 4.1 TOPKAPI-ETH

#### 4.1.1 Model description

TOPKAPI-ETH is a physically-oriented, fully distributed, glacio-hydrological model that was adapted from a rainfall-runoff model (Ciarapica and Todini, 2002) to simulate snow cover evolution and glacier mass balance in high-mountain areas. The model has been used successfully in the semiarid Andes (Ragettli et al., 2014; Ayala et al., 2016), the Alps (Fatichi et al., 2014, 2015) and the Himalaya (Ragettli et al., 2013, 2015), and it is well-suited for long-term simulations (Ragettli et al., 2016).

TOPKAPI-ETH is forced with time series of precipitation, air temperature and cloud transmissivity of solar radiation. The model simulates snowfall at a given grid cell when precipitation occurs and air temperature is below a threshold parameter. When snow accumulation exceeds a slope-dependent threshold of a given grid cell (snow holding depth, $S_{hd}$), excess snow is moved to a lower grid cell based on the SnowSlide gravitational transport model (Bernhardt and Schulz, 2010):

$$S_{hd} = SGR_C \cdot e^{SGR_a \cdot SLP}, \qquad (2),$$

where $SGR_C$ (m) and $SGR_a$ are empirical parameters and $SLP$ is the slope of the grid cell. Snow and ice melt is calculated with the Enhanced Temperature-Index (ETI) model (Pellicciotti et al., 2005), depending on the net solar radiation and near-surface air temperature:

$$M = \begin{cases} SRF \cdot S_{in} \cdot (1 - \alpha) + TF \cdot T_a, & T_a > T_T \\ 0, & T_a \leq T_T \end{cases}, \qquad (3),$$

where $M$ is melt (mm h$^{-1}$), $SRF$ is the shortwave radiation factor (mm m$^2$ h$^{-1}$ W$^{-1}$), $S_{in}$ is the incoming shortwave radiation (W

m$^{-2}$), $\alpha$ is surface albedo, $TF$ is the temperature factor (mm h$^{-1}$ °C), $T_a$ is air temperature (°C), and $T_T$ is the air temperature threshold parameter for the onset of melt (°C). TOPKAPI-ETH internally converts the units of the ETI variables and parameters to a daily time step. To calculate ice melt under supra-glacial debris we also use the ETI model but with reduced melt factors (see section 4.1.3). Although TOPKAPI-ETH includes a melt module that accounts for debris thickness in the computation of sub-debris ice melt, we did not to use it due to the lack of debris thickness information in the region, and the large uncertainties

that are present in large-scale estimates of debris thickness (Rounce and McKinney, 2014; Schauwecker et al., 2015).



Once snow accumulation and melt are integrated to calculate the annual glacier surface mass balance, TOPKAPI-ETH translates it to elevation changes at the end of each hydrological year (from April to March) by means of the Δh-approach (Huss et al., 2010). This is done by using the originally-proposed, glacier-size dependent parameters (cf. Fig 3b in Huss et al. (2010)). At the end of March, if the annual mass balance was negative the model performs a reduction of glacier area, but no

area increases due to positive mass balances are prescribed. While snow melt over a non-glacierized grid cell is added to the respective soil layers, snow and ice melt over glaciers are added to a conceptual water reservoir for each glacier, which releases its water by means of a linear reservoir equation (Jansson et al., 2003). In non-glacierized grid cells, the model simulates subsurface water flow, evapotranspiration, and water routing (Ciarapica and Todini, 2002).

### 4.1.2 Model setup for the Maipo River Basin

We setup an instance of the TOPKAPI-ETH model for the entire Maipo River Basin at a spatial resolution of 1 km that does not include glaciers. Glaciers and their runoff contribution are accounted for separately in the next section, but their ice melt contribution is included in this section for the calibration of the sub-surface parameters. The objective of the 1-km resolution setup for the entire Maipo River Basin is to simulate snowmelt and rain, which account for the largest runoff volumes in the basin, at a resolution that allows for multiple model runs and the automatic calibration of the sub-surface flow parameters. The

model is run continuously from 1955 to 2016 at a daily time step.

We spatially distribute daily precipitation over the basin using monthly mean maps derived from the CR2 precipitation product. The spatial distribution is made from basin-averaged precipitation extracted from the CAMELS-CL database (Álvarez-Garretón et al., 2018) for the period 1979-2016, and from Quintal Normal station for the period 1955-1978. Daily mean air temperature is extrapolated from Embalse El Yeso using basin-scale daily temperature lapse rates (see section 3.3). Periods

with no direct information of daily mean air temperature at Embalse El Yeso are filled using correlation with records of daily extreme temperatures at the same station (mainly the period 1962-1977) or at Quinta Normal station (1955-1979).

The calibration of the Maipo River Basin model consists of two steps: (i) the snow parameters are varied in order to fit basin-scale SCA and SWE against the MODIS and CAMELS-CL products (section 3.4), and (ii) the parameters controlling sub-surface fluxes are varied in order to fit monthly mean streamflow records at Maipo en El Manzano. While parameters in step

(i) are manually calibrated and largely correspond to default values from previous studies using TOPKAPI-ETH, parameters in step ii) are automatically calibrated minimizing three different evaluation metrics (Nash-Sutcliffe (NS), Root Mean Squared Error (RMSE), and mean bias (BIAS)).

During the calibration procedure, we find that the use of the precipitation amounts derived from the CR2 product leads to an underestimation of SCA and SWE over the basin area, and streamflow at the basin outlet. This underestimation of precipitation

by the CR2 product was already identified by Álvarez-Garretón et al. (2018) when analysing runoff ratios across Chile, and attributed to a limitation of satellite-derived precipitation estimates over high-elevation areas. Similar results have been found



in this region using a regional climate model driven by ERA-Interim (Bozkurt et al., 2019), and the MERRA reanalysis (Cortés et al., 2016). Although the CR2 precipitation product corrects the ERA-Interim values by contrasting them with ground data, these data are available only below 3000 m a.s.l. in this region. In our study, to fit the seasonal average curves of SCA and

SWE, and close the water balance of the basin, we correct the precipitation derived from the CR2 product by 50%. This correction generates precipitation amounts in the order of 3 to 4 times larger than that registered on low-lying areas. This value is larger than those estimated by previous studies on the west side of the semiarid Andes (Falvey and Garreaud, 2007; Viale et al., 2011; Cortés et al., 2016), which estimated that the orographic effect results in a precipitation enhancement in the order of 2 to 3. The spread of precipitation amounts estimates over the semiarid Andes (and in general over mountain areas) is in fact

large, and previous hydrological studies have performed different types of corrections to close the water balance at the basins' scale (Vicuña et al., 2011; Ragettli and Pellicciotti, 2012; Burger et al., 2018a).

An additional problem identified during the model calibration is that air temperature over areas above 5000 m a.s.l. (about 5% of the basin) is most of the time lower than the air temperature threshold parameter for melt onset, generating large snow accumulation that is not seen in the SWE reconstruction product. As snow on this high-elevation areas is in reality removed

by wind transport and sublimation, we reset the SWE in the model to zero at the beginning of each hydrological year. Although this implies that the model is not strictly mass-conserving, we verify that the discarded snow is in average 34 mm yr$^{-1}$ over the entire basin (or 688 mm yr$^{-1}$ = 1.9 mm d$^{-1}$ over the areas above 5000 m a.s.l.), which is a reasonable estimate of sublimation amounts for this region (Corripio, 2003; Ayala et al., 2017b), and is in the order of the model uncertainties (see Figure 2).

Figure 2 shows the results of the model calibration for daily time series of SWE (Figure 2a), monthly time series of streamflow

(Figure 2b), and seasonal variations of SCA (Figure 2c), SWE (Figure 2c) and streamflow (Figure 2d). The final calibrated snow parameters for this setup are shown in Table 1, whereas values for the sub-surface flux parameters are shown in the supplementary information (Table S1). The quality metrics for snow and streamflow variables show very good results in both the calibration and validation periods. Extreme values are well captured, except for the humid winter of 1988, in which the model underestimates snow accumulation and streamflow.

**4.1.3 Model setup for individual glaciers**

In addition to the basin-scale model, we set up an instance of TOPKAPI-ETH for each one of the glaciers larger than 1 km$^2$ in the catchment (about 59 glaciers). These instances have a spatial resolution of 100 m, which is more adequate to simulate the processes governing glacier mass balance. The domain of these models runs correspond approximatively to the smallest catchment that contains the 1955 glacier extent of each glacier. The models are run at a daily time step starting in the year

1955, and are then re-started in 2000 using the topographic and geographic information from that year. The models are forced using daily precipitation at the location of the centroid of each glacier, linearly interpolated from basin-averaged precipitation (including the 50% precipitation correction) (Álvarez-Garretón et al., 2018), and assumed uniform over each corresponding





domain. Air temperature is extrapolated from the Embalse El Yeso meteorological station using a constant air temperature gradient equal to the environmental lapse rate (-6.5 °C km⁻¹). For the study period in which no CR2 precipitation products are

available, the Quinta Normal and Embalse El Yeso stations are used.

We choose a set of model parameters typically used in the literature for this region (Ragettli and Pellicciotti, 2012; Ayala et al., 2016; Burger et al., 2018a) for all individual glacier models and keep parameter calibration at a minimum level. For each glacier, we vary only the ETI model parameters within ranges suggested in the literature (Finger et al., 2011; Ragettli and Pellicciotti, 2012; Ayala et al., 2017a) to fit the glacier-wide mass balance as derived from the geodetic mass balances. A

summary of literature-derived and calibrated parameters for the individual models is shown in Table 1. Within each model, melt factors for debris-covered areas are fixed to 25% of the values for debris-free areas. The 25% factor is estimated from the comparison between melt rates on debris-free and debris-cover sites on Piramide, Bello and Yeso glaciers in the Estero del Yeso catchment (Ayala et al., 2016; Burger et al., 2018a), a sub-catchment of the Maipo River Basin.

Although we set up a TOPKAPI-ETH model for all glaciers with an area above 1 km² in 2000 (equivalent to 59 glaciers), we

find that staying within the selected ranges for the ETI parameters only allows to fit the geodetic mass balances in 26 cases. Among the discarded glaciers, about half of them are smaller than 3 km², and the rest correspond to those lying on the slopes of the Tupugatito Volcano and San José volcanic complex (Volcán Tupungatito, Azufre, and Marmolejo glaciers). We suspect that this is an expression of the fact that some of the processes not included in TOPKAPI-ETH (namely permafrost, sublimation, snow dynamics or geothermal fluxes) may play a role governing the mass balance of these glaciers. However, it

might also be related to local deficiencies in the spatial distribution of air temperature and precipitation. No rock glaciers are included in this subset of glaciers. The location and main properties of the 26 modelled glaciers in comparison with those of the total sample are shown in Figure 3 and Table 2, respectively. The simulated glaciers are spread over the entire basin, and their mean elevations are in the middle range of the total sample. Glaciers smaller than 1 km², from which 85% correspond to rock glaciers or glacierets, are less-well represented by the sample of 26 glaciers. The sample of 26 glaciers is mostly oriented

towards south (aspect > 90°) and does not include the steepest glaciers. In Figure 3, we also highlight the areas with the discarded large glaciers on Tupungatito Volcano and San José volcanic complex.

The results of the calibration of the TOPKAPI-ETH models for the 26 modelled glaciers are shown in Figures 4a (period 1955-2000) and 4b (2000-2013). The calibration results are very good for both periods with area-weighted RMSEs of 1 and 0.2 m w.e. for the 1955-2000 and 2000-2013 periods, respectively. These errors are well within the uncertainty bounds of the geodetic

mass balance. Figure 4c shows the resulting cumulative glacier mass balance for all simulated glaciers, their area-weighted average, and the comparison with the glaciological mass balance measured on the Echaurren Norte Glacier since 1975. The fastest declining line of the sample corresponds to the Olivares Alfa Glacier, which has been previously identified as one of the glaciers with the largest retreating rates in the basin (Malmros et al., 2016). Interestingly, several of the glaciers show a





positive or near-neutral mass balance over the entire period, which might be an indication that these glaciers have already

retreated close to a new equilibrium.

## 4.2 Extrapolation

We extrapolate the mass balance of the 26 modelled glaciers to the entire basin based on the methodology described by Huss (2012). In that work, a set of in-situ glacier mass balance measurements for Switzerland were used to calculate the mass balance of all glaciers in the European Alps. Here, we calculate the surface mass balance $B$ (m w.e.) of glacier $g$ in year $y$ with:

$$B(g,y) = \bar{B}(g,p) + \Delta B(s,y), \qquad\qquad (4),$$

where $\bar{B}(g,p)$ is the average annual mass balance in the study period $p$, and $\Delta B(s,y)$ is the glacier mass balance anomaly in the sub-catchment $s$, where glacier $g$ is located. While $\bar{B}(g,p)$ is extracted from the geodetic mass balance, $\Delta B(s,y)$ is derived from the TOPKAPI-ETH simulations. Equation (4) is applied to the periods 1955-2000 and 2000-2013 by calculating $\bar{B}(g,p)$ from the geodetic mass balance. The term $\Delta B(s,y)$ is calculated as the anomaly of annual mass balance of simulated

glaciers located in the sub-catchment $s$ for each study period, i.e.:

$$\Delta B(s,y) = B(g^{*,s},y) - \bar{B}(g^{*,s},p), \qquad\qquad (5),$$

where $g^{*,s}$ is the subset of modelled glaciers ($*$) in sub-catchment $s$.

The time series of annual mass balance $B(g,y)$ are then used to estimate the volume changes of each glacier throughout the study periods. Glacier areas ($S$) are updated due to negative changes in glacier volume ($V$) (we do not prescribe increases of

glacier area due to positive annual mass balance) by means of the area-volume scaling formula:

$$S = \left(\frac{V}{c}\right)^{\frac{1}{\gamma}}, \qquad\qquad (6),$$

where $\gamma$ and $c$ are the scaling parameters. In line with recommendations of the volume-area scaling theory (Bahr et al., 2015), the parameter $\gamma$ is kept constant in all periods at a value of 1.357, and we let $c$ to vary in order to fit the total glacier volume in the basin in years 1955 and 2000 (calculated in section 3.2). Parameter $c$ is calculated as 28.1 for 2000 (this value is also

used afterwards), but a value of 21.1 is the one that fits best to our estimates of ice thickness in 1955. In between these two years we use a linear interpolation of $c$.

In the calculation of area and volume evolution, we account for the uncertainties in the annual mass balance, inventoried glacier areas in 1955 and 2000, and the parameter $c$, by disturbing each variable with a random variation. These random variations are 1000 realizations of three normal probability distributions of mean 0 and standard deviations equivalent to the typical errors

of each variable. From the uncertainties in the geodetic mass balance, we estimate a typical error in the annual mass balance of 0.08 m w.e. yr$^{-1}$ for the period 1955-2000 and 0.13 m w.e. yr$^{-1}$ for 2000-2016. Based on Paul et al. (2013), we assign a 5%





error to the area of each glacier in the year 2000 inventory, and we double this value for the 1955 inventory. The error for the parameter c is calculated in order to match the uncertainty in our ice thickness estimates, and results in a value of 4.1.

Glacier runoff, including all its components (i.e. ice melt, snow melt and rain), is extrapolated directly from the TOPKAPI-
ETH results for the 26 modelled glaciers to the rest of the glacierized areas, and do not depend on the extrapolated time series of glacier mass balance. The uncertainty in glacier runoff is estimated at each year as proportional to that calculated for glacier volume. As in Huss and Hock (2018), we define glacier runoff as the water originating from the initially glacierized area (1955 in our case), i.e. independent of the glacier area in a particular year. This allows the evaluation of changes in total headwater runoff due to glacier retreat. However, in our study we also evaluate specific variations of the ice melt component. Throughout
the manuscript, glacier runoff and its components are presented as normalized by the area of the entire Maipo River Basin.

**4.3 Committed ice loss estimates**

We estimate the committed glacier ice loss caused by the temperature increase in the last decades by conducting a set of ten additional TOPKAPI-ETH simulations, and by extrapolating them using the same analysis as described in the previous section. The additional simulations are run under different synthetic climate scenarios in which the climate of the last two decades is
stochastically repeated for a 100-year period. The meteorological inputs are built by repeating 1-year long blocks of the input variables (precipitation, temperature and cloud transmissivity) corresponding to a randomly selected year between 1993 and 2016 (23 years). We select this period because air temperature was relatively stable in the basin, and precipitation showed the characteristic inter-annual variability of this region.

While the anomaly term $\Delta B(s, y)$ is calculated in the same way as for the period 1955-2016 (i.e. from the TOPKAPI-ETH
simulations), as no geodetic mass balances are available for the synthetic scenarios, we calculate $\bar{B}(g, p)$ using two different approximations depending on glacier size. For glaciers that are larger than the size of the smallest modelled glacier (1.1 km$^2$), we use a multiple linear regression of the mass balance of the modelled glaciers in each scenario with their topographic parameters in year 2000:

$$\bar{B}(g, p) = a_1 \cdot x_1 + \cdots + a_n \cdot x_n \qquad (7),$$

where $a_i$ are calibrated coefficients and $x_n$ are topographic parameters. In average for the 10 synthetic scenarios, the best results are given by glacier area, median glacier elevation, percentage of debris cover, mean sky view factor, and mean aspect. Together, these five variables explain 52% of the total variance, which is in the range of the original application of this methodology (Huss (2012) obtained 35% using three variables and 51% using six). Results of this procedure are summarized in Table 3. For glaciers smaller than 1.1km$^2$, we use the average mass balance of modelled glaciers in the corresponding sub-
catchment. As in the 1955-2016 period, rock glaciers are assumed to have a balanced mass budget. Once the time series of mass balance for the 10 synthetic scenarios are calculated, we compute area and volume evolution of each glacier, and their associated uncertainties, using the same methodology as for the 1955-2016 period.



## 5 Results

Figure 5 and Table S2 (in the supplementary information) present a summary of the simulations in this study. In the period
1955-2016, we conduct the TOPKAPI-ETH simulations for the Maipo River Basin (SIM-1A) and the 26 modelled glaciers
(SIM-1B), and the extrapolation for all glaciers (SIM-1C). Using the synthetic meteorological time series derived to calculate
the committed ice loss, we conduct 10 additional TOPKAPI-ETH simulations for the Maipo River Basin (SIM-2A) and the
modelled glaciers (SIM-2B), which are then also used for extrapolation (SIM-2C).

### 5.1 Glacier changes and runoff contribution in the period 1955-2016

In Figure 6, we present the temporal variability of precipitation (a), air temperature (a), the equilibrium line altitude (ELA) (b)
and cumulative mass balance (c and d) in the Maipo River Basin since 1955. While the large inter-annual variability of the
basin's mean precipitation (Figure 6a, blue bars) directly relates to the El Niño Southern Oscillation (ENSO) phenomenon, a
3-year moving average of this variable exposes a sequence of dry (e.g. 1967-1969, 2010-2016) and wet periods (e.g. 1978-
1987 and 2000-2008). This sequence has been related to other climatic indices, such as the Pacific Decadal Oscillation (PDO)
or the Interdecadal Pacific Oscillation (IPO) (Boisier et al., 2016; González-Reyes et al., 2017). From 2010 on, precipitation
has decreased due to a severe drought across Chile (Garreaud et al., 2017). Air temperature over the basin shows a sustained
increase in the long-term, but with relatively stable values since the mid-1990s. Since the 1960s, air temperature has increased
in about 2°C. Figure 6b shows the annual and decadal variability of the ELA of the 26 modelled glaciers. The ELA is calculated
as the average elevation of all grid cells with an annual mass balance of ±10 cm, and the estimated range (in light red)
corresponds to the standard deviation. Since the 1960s, the elevation of the ELA has increased by 370 m, or 66 m per decade.

Figures 6c and 6d integrate the results of TOPKAPI-ETH and the extrapolation procedure. Figure 6c presents the cumulative
surface mass balance of glaciers in the Maipo River Basin since 1955, including the 26 glaciers modelled with TOPKAPI-
ETH, and the remaining glaciers in the basin for which extrapolation was used. The cumulative mass balance shows a
decreasing trend interrupted by short periods of positive or near-neutral mass balance, with a more negative final value for the
26 modelled glaciers than for all glaciers in the basin. The more negative value for modelled glaciers might be caused by their
larger area in comparison to the rest of the glaciers, as large glaciers have shrunk more extensively (Malmros et al., 2016). For
comparison with the long-term glacier mass balance reference in the region, we include the direct measurements on Echaurren
Norte Glacier, which presents a more negative trend, most likely due to its low elevation (3650 to 3900 m a.s.l.). In Figure 6d,
we present the surface mass balance of glaciers in each sub-catchment, where relatively large differences can be seen. In
general, glaciers in southern catchments show more positive mass balance than those in northern catchments. Most notably,
glaciers in Olivares show the most negative mass balance throughout the study period, whereas those in Volcán present a
positive mass balance until the mid-2000s. However, after the start of the current drought in 2010, negative glacier mass
balances dominate across the entire Maipo River Basin.





In Figure 7 we show the variations of glacier runoff and its components (ice melt, snow melt and rain) in the initially glacierized
areas over the period 1955-2016. While the annual and summer inter-annual variability is presented in Figure 7a and 7b, Figure
7c presents the average seasonal curve and the percentage of each contribution. The summer period is chosen as January to
March. Glacier runoff was 186±27 mm yr$^{-1}$ over the entire period and shows a sequence of three decreasing maxima (1968-
1969, mid-1980s, and end of 2000s). Glacier runoff peaked at 257±64 mm yr$^{-1}$ during the severe drought of 1968-1969 (the
driest hydrological year in record) and it averages 166±30 mm yr$^{-1}$ during the current drought (2010-2016). Figure 7a shows
that the inter-annual variability of ice melt is very large (with a coefficient of variation of 0.57), and its share in total glacier
runoff can vary from less than 10% (as in 1982-1983, 1997-1998, and 2002-2003) to more than 90% (as in 1968-1969). Except
for 1968-1969, snow melt on this areas is consistently the largest runoff contributor at the annual scale, but the contribution
during summer is very variable. In Figure 7c, we show the summary of runoff contributions at the annual scale. Runoff
contribution is dominated by snowmelt (60%), with ice melt representing 37% of the annual total. Rain represents about 3%,
but these amounts have increased since 1955 (Figures 7a and 7b).

In Figure 8, we quantify the role that glacier runoff has played in the entire Maipo River Basin over the study period. At the
annual scale, glaciers provide 17±7% of the total runoff, but this contribution can increase up to 60±23% in summer. In 1968-
1969, the runoff contribution from the 1955 glacier areas provided 51% of the annual runoff, and almost 100% during summer.
During the current drought, glacier runoff has represented 17% of the annual runoff and 57% of summer runoff. The value of
17% during the current drought is close to the average value over the entire study period.

**5.2 Glacier changes and runoff contribution for the committed ice loss scenarios**

Figure 9 presents the evolution of glacier volume (a), area (b), and runoff (c) in the Maipo River Basin, in the past period
(1955-2016) and the committed ice loss scenarios. To assess the changes of glacier area and volume we use the values estimated
for the year 2000 as reference, whereas for glacier runoff we use the average in the period 1955-2016. As mentioned above,
the committed ice scenarios do not represent a realistic projection for the future, and we use the years of 2000 to 2100 in the
x-axis for visual purposes only. Glacier area and volume varied by –38±5% (from 558±56 to 347±27 km$^2$) and –20±14% (from
18.6±4.5 to 14.9±2.9 km$^3$) in the period 1955-2016, respectively and if glaciers were in equilibrium with the current climate,
the glacier area and volume would reduce to 79±18% and 81±38% of the 2000 values, respectively. The uncertainty in glacier
area and volume derived from our calculations (blue bands) reduces from 2000 on, due to the higher accuracy that we assign
to the year 2000 inventory and DEM. In the committed ice loss scenarios, uncertainty starts on similar levels as that in 2000,
but as the scenarios differentiate from each other, the uncertainty increases towards the end of the simulation period. In Figure
9c presents glacier runoff from the initially glacierized areas normalized by the area of the Maipo River Basin. Glacier runoff
in the committed ice loss scenario decreases quickly until a relatively steady value is reached at 79±33% of the average glacier
runoff in the 1955-2016 period. This value is equivalent to 61±24% of that in the 1968-1969. Uncertainty bounds in Figure 9c
are proportional to those of the glacier volume.





As the large precipitation inter-annual and inter-decadal variability could mask the runoff trends associated only with the reduction of glacier volume, in Figure 10 we present the variability of ice melt in the period 1955-2016 and the committed ice loss scenarios. For this figure, we use the maximum value of ice melt in the period 1955-2016 as reference, which corresponds to the hydrological year 1968-1969. Although also ice melt shows a very large inter-annual variability, it is clear that the

maximum values have decreased over the last decades (see maximum values in 1968-1969, 1990-1991, and 2011-2012). We estimate that if glaciers reached equilibrium with the current climate, the peaks would be considerably lower than those in the 1955-2016 period. In this equilibrium situation, the peaks are close to 40% of the largest ice melt runoff contribution in the past (1968-1969).

## 6 Discussion

### 6.1 Glacier changes

Our results indicate that the total glacier volume in the Maipo River Basin decreased in about one fifth in the period 1955-2016. The cumulative glacier mass balance in the Maipo River Basin shows variations that are similar to those registered on Echaurren Norte Glacier. These variations consists of a general decreasing trend, concurrent with an increase of the ELA (Carrasco et al., 2005, 2008), which has been interrupted by periods of slightly positive or neutral mass balance. Since the mid-

1980s, there has been a strong mass loss, interrupted only by a positive period in the beginning of the 2000s. In fact, from 2000 on, we observe a 10-year period with positive or nearly neutral mass balance. This was also described by glaciological observations (Masiokas et al., 2016) and geodetic mass balances (Braun et al., 2019; Dussaillant et al., 2019). Following this period, strongly negative mass balances have been observed (Masiokas et al., 2016; Burger et al., 2018a), concurrent with a severe drought in central Chile, unprecedented in extension and duration (Garreaud et al., 2017). For the period before 1975,

when the mass balance measurements on Echaurren Norte Glacier started, we compare our results to the reconstruction obtained by Masiokas et al. (2016). The latter estimated a strongly negative mass balance in the period 1955-1975, whilst we obtain a nearly neutral mass balance between 1955 and 1968, and a more negative balance from 1968 to 1975. These differences might either correspond to differences between the basin-averaged mass balance and that on Echaurren Norte Glacier, or be a consequence of the different methodologies between our study and that of Masiokas et al. (2016). While

Masiokas et al. (2016) relies on the correlation between hydro-meteorological records and the measured surface mass balance on Echaurren Norte, our model is calibrated to the geodetic mass balances.

The different trends of glacier mass balance in the sub-catchments of the Maipo River Basin are an expression of the diverse climatic and morphological characteristics that dominate across the basin. For example, the positive and near-neutral glacier mass balances in Volcán and Upper Maipo might be related to higher precipitation towards the south, or that several glaciers

have retreated close to a new equilibrium state. Within the Olivares sub-catchment, the geodetic mass balance is in line with the large areal changes found by Malmros et al. (2016). This might be explained by a strong imbalance of the large glaciers in





that catchment, and/or by the impacts of nearby mining activities (Los Bronces and Andina mines), especially dust deposition on Olivares Alfa Glacier and its neighbour glaciers. However, more specific studies addressing albedo changes are necessary to obtain more conclusive results.

Our estimates of committed ice loss show that glaciers will continue to shrink if the climate remains stable, with an estimated committed ice loss of 20% relative to the volume in the year 2000 (30% relative to 1955). We stress that these estimates do not correspond to a realistic future scenario, but are an indication of the glacier changes that past climate will produce in any case. Future projections under emission scenarios will certainly show more dramatic reductions of glacier area and volume. In the context of future projections, we highlight that most projections for glacier changes in the Central Andes (Marzeion et al.,
2012; Radić et al., 2014; Huss and Hock, 2015) are included in the macro-region of the "Southern Andes", which also contains the Patagonian Ice Fields. Future projections of glacier changes in the region are thus strongly influenced by these large ice masses, with their peculiar climate, and physical processes (e.g. calving) that are not representative of the small mountain glaciers along the semiarid Andes (Mernild et al., 2015).

### 6.2 Glacier runoff

Despite the reduction of glacier volume in the period 1955-2016, our estimates of glacier runoff do not show (Figure 8 and 9c) the typical increasing or decreasing phases of peak water observed or projected for other catchments across the world (Baraer et al., 2012; Farinotti et al., 2012). This result is similar to that obtained by Casassa et al. (2009), who did not find significant trends in an analysis of Maipo streamflow records. Peaks in glacier runoff have reduced their magnitude over the last decades as a combination of a decrease in precipitation (Boisier et al., 2016) and the reduction of ice volume, but it is difficult to
identify if there was an increasing phase of glacier runoff in the period 1955-2016.

We suggest that the strong inter-annual and inter-decadal climatic variability observed in the semiarid Andes (Montecinos and Aceituno, 2003; Masiokas et al., 2006; Falvey and Garreaud, 2007) is also transferred to the glacier runoff time series, modifying or masking the typical trends associated with glacier retreat. Once an extended time period is considered (in this case a committed ice loss scenario, Figure 9c), peak water emerges more clearly. Huss and Hock (2018) estimated that glacier
runoff in the Rapel River Basin (south of the Maipo River Basin) experiences peak water in the current decade (2010-2020), but their analyses also show a strong inter-annual glacier runoff variability from 1980 to 2010. This makes peak water evident only when compared to the future projections under different emission scenarios, in which the glacier runoff is considerably lower than present levels.



### 6.3 Uncertainties in the modelling of glacier changes in data-scarce regions

We identify four main sources of errors and uncertainties in our study: (i) the glaciological datasets, i.e. the geodetic mass balance, glacier outlines, ice thickness and debris cover areas, (ii) the spatial distribution of meteorological inputs, (iii) modelling limitations in TOPKAPI-ETH, and (iv) limitations of the extrapolation methodology.

In general, the uncertainties of the elevation changes and glacier properties are well quantified and explicitly stated in the confidence bounds of the geodetic mass balance (Figure 9). However, a few properties were not explicitly quantified, such as
the ice content in rock glaciers, which is in fact a key problem in the semiarid Andes (Schaffer et al., 2019). Due to this, there might be an overestimation in the ice content due to the presence of rock glaciers. In the future, more geophysical measurements to acquire information on ice content in rock glaciers of the semiarid Andes (e.g. Croce and Milana, 2002) could improve the estimates of runoff generation from these landforms.

The accuracy in the spatial distribution of meteorological inputs is particularly difficult to evaluate, and it likely corresponds
to a major source of uncertainty, especially precipitation. This is because of the relatively sparse network of meteorological stations installed in the basin, the difficulties of atmospheric models to represent precipitation processes over the Andes (Bozkurt et al., 2019), and the underestimation of satellite-based precipitation products over high-elevation areas (Álvarez-Garretón et al., 2018). However, the indirect evaluation of precipitation amounts through snow cover products and the basin's water balance increase the confidence in the results of this study.

In relation to TOPKAPI-ETH, it has been shown that the parameterizations of snow accumulation and ablation included in TOPKAPI-ETH work well for wind-sheltered locations (Ayala et al., 2017a). However, the representation of processes driving the mass balance at some specific sites requires more fundamental work, and additional parameterizations or more physically-based representations are required. Such sites correspond mainly to sublimation-dominated sites above 5500 m a.s.l., where the temperature-index modelling is inaccurate (Ayala et al., 2017a, 2017b), debris-covered areas with complex distributions
of debris thickness (Burger et al., 2018a), or steep glacierized slopes such as the volcanoes in the Maipo River Basin.

As the average rates of annual mass balance in the 1955-2016 period are calculated from the geodetic mass balance, the long-term glacier changes derived from the extrapolation methodology should be well simulated, but the year-to-year variations of mass balance depends on the representativity of the modelled glaciers. As discussed by Huss (2012), a low representativity of the reference glaciers could lead to large errors in the mass balance of individual glaciers and years, but these errors should be
lower at the mountain-range scale and over long time periods. In the committed ice loss scenarios, the uncertainty of glacier changes is higher because it also relies on the multiple linear regression analysis. In any case, we explicitly accounted for the uncertainties in the extrapolated mass balance (are least partly) by means of the random perturbation described in section 4.2.



## 7 Conclusions

We have reconstructed the changes that glaciers in the Maipo River Basin experienced over the last six decades, with a focus
on glacier runoff and the impacts of its long-term variations on the basin's hydrology. These results add a missing piece to the
current hydro-climatological knowledge of the semiarid Andes, and can be useful for water managers and stakeholders to
develop adaptation or mitigation strategies. Although some uncertainties still remain, our results successfully take into account
a number of independent datasets, including snow cover area variations, snow water equivalent reconstructions, streamflow
records, glacier inventories and geodetic mass balances.

Our main conclusions are as follows:

a. Over the period 1955-2016, the total glacier volume in the Maipo River Basin has decreased by 20±14% (from
18.6±4.5 to 14.9±2.9 km$^3$), respectively. In agreement with other studies, our results show that the cumulative glacier
mass balance over the study period had a general decreasing trend, interrupted by short periods of positive or near-
neutral mass balance. This might be an indication that some glaciers temporarily retreated to a new equilibrium state.
Strongly negative mass balance have dominated since the start of the current drought in 2010. Despite the general
trend, there are important differences between the glacier mass balances of the sub-catchments, with the southern sub-
catchments (Volcán and Upper Maipo) showing positive or near-neutral mass balances until 2000, and the Olivares
sub-catchment showing a strongly negative mass balance over the entire period.

b. The average glacier contribution to runoff in the Maipo River Basin – i.e. the runoff contribution from liquid
precipitation, snowmelt and icemelt from the areas that were glacierized in 1955 – was 186±27 mm yr$^{-1}$ in the period
1955-2016. Instead of a clear peak water, we identify a decreasing sequence of runoff maxima that can be linked to
both a decrease in precipitation since the 1980s and a reduction of ice melt. Glacier runoff has decreased since the
severe drought of 1968-1969, when glacier runoff peaked at 257±64 mm yr$^{-1}$ (51% of the basin's total runoff). During
the current drought, which started in 2010, the contribution was 166±30 mm yr$^{-1}$ (17% of the total runoff).

c. If climate was to stabilize at the level of the past two decades, we estimate a committed glacier ice mass loss of
19±38%. This would cause glacier runoff to reduce by 21±33% when compared to the 1955-2016 average, or by
39±24% when compared to 1968-1969. Based on these numbers, we anticipate that the future capacity of the basin
to mitigate severe droughts will be reduced.

Our results shed light on the glacier runoff evolution in the semiarid Andes, and complement recent studies that assessed
regional-scale glacier changes (Braun et al., 2019; Dussaillant et al., 2019). Some topics deserving further attention that should
be addressed are the drivers behind the positive mass balance in the southern catchments (Volcán and Upper Maipo), the
processes governing mass balance on glaciers on active volcanoes, the possible anthropogenic impacts on glaciers in the
Olivares sub-catchment, and the quantification of the hydrological role of rock glaciers. Whilst our simulations of committed
ice loss provide estimates of the minimum changes that glaciers will experience due to past changes of the climate, future





studies driven by climate model simulations and emission scenarios should provide more realistic projections for the future of the region's glaciers.

**Data availability**

Data used in this study are available upon request to the authors.

**Author contributions**

AA designed the study with contributions from DF and DFB, and performed the calculations with TOPKAPI-ETH and the extrapolation method. DFB calculated the geodetic mass balances. MH calculated the ice thicknesses for 2000. All co-authors provided scientific advice during the study. AA prepared the manuscript with contributions from all co-authors.

**Competing interests**

The authors declare that they have no conflict of interest.

**Acknowledgements**

AA acknowledges Shelley MacDonell for support at CEAZA and Eduardo Muñoz for his help in the calibration of TOPKAPI-ETH. The work benefited from the financial support of seed grant within ETH Zurich's Research for Development (R4D) funding scheme, and was conducted within the research collaboration *"Glacier runoff contributions in the Maipo River catchment"* between the Fundación para la Transferencia Tecnológica (UNTEC), Santiago, Chile and ETH Zurich,
Switzerland.

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





**Figure 1: a)** Maipo River Basin next to the city of Santiago, in central Chile; **(b)** the basin outlet and the sub-catchments, rivers, glaciers, and hydro-meteorological stations; **(c)** the elevation range of every glacier in the basin as a function of the average latitude (arbitrary scale) in each sub-catchment, and the mean elevation (black line); **(d)** estimated total ice volume (left axis), and glacierized area (right axis) in each sub-catchment. The surface and glacier type (debris-free, debris-covered or rock glacier), as well as the number of glaciers in each sub-catchment are indicated.
**Figure 2: Results of the calibration of the TOPKAPI-ETH model for the Maipo River Basin. (a) Simulated SWE against results of Cortés et al. (2016), (b) Simulated and observed monthly streamflow at the basin outlet. In (a) and (b) the light orange area indicates the calibration period. (c) Average seasonal variability of simulated and observed SCA from Aqua and Terra missions and SWE from Cortés et al. (2016) in the calibration period, (d) average seasonal variability of simulated and observed streamflow in the calibration period. The coloured areas in (d) correspond to the observed and simulated standard deviations from the inter-annual variability. Model metrics are indicated for the calibration and validation periods and correspond to NS: Nash-Sutcliffe coefficient, RMSE: Root Mean Square Error, and BIAS: average bias.**



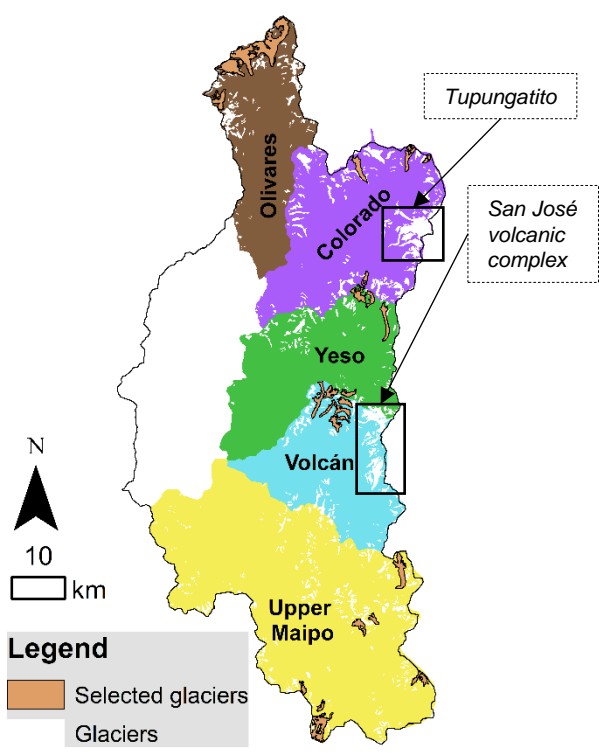

**Figure 3: Location of the 26 glaciers modelled with TOPKAPI-ETH. We highlight the volcanic areas on which some large glaciers were discarded from the modelled sample.**





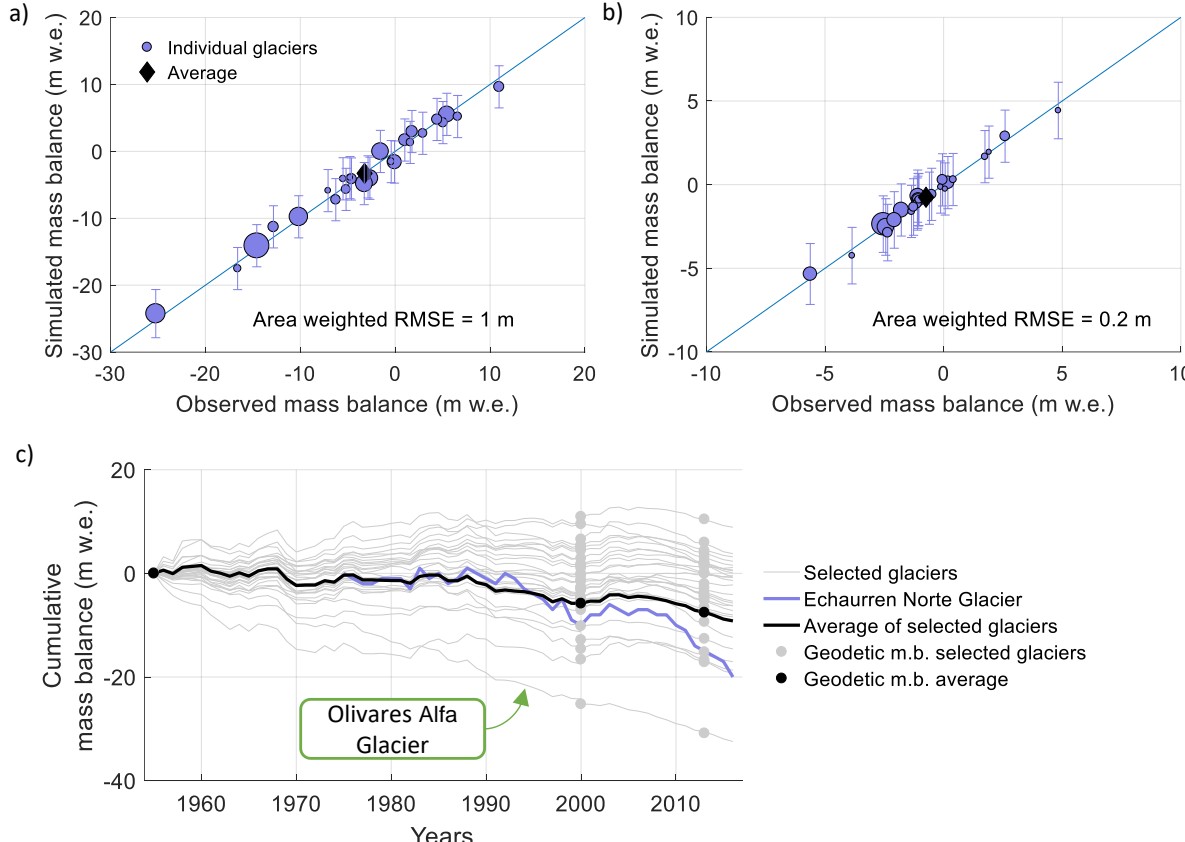

815

**Figure 4: Results of the calibration for the 26 modelled glaciers. Glacier-averaged mass balance (blue circles) simulated with TOPKAPI-ETH and observed from geodetic mass balances in the period (a) 1955-2000 and (b) 2000-2013. The area-weighted average of all glaciers is indicated with black diamonds. The blue bars show the uncertainty of the geodetic mass balances, (c) Cumulative 1955-2016 mass balance for each modelled glacier (grey lines), the area-weighted average of all glaciers (black line) and the mass balance measured on Echaurren Norte Glacier (blue line). The geodetic mass balances for each modelled glacier is indicated with a circle. The curve corresponding to the fastest retreating glacier of the sample, Olivares Alfa Glacier, is labelled.**

820



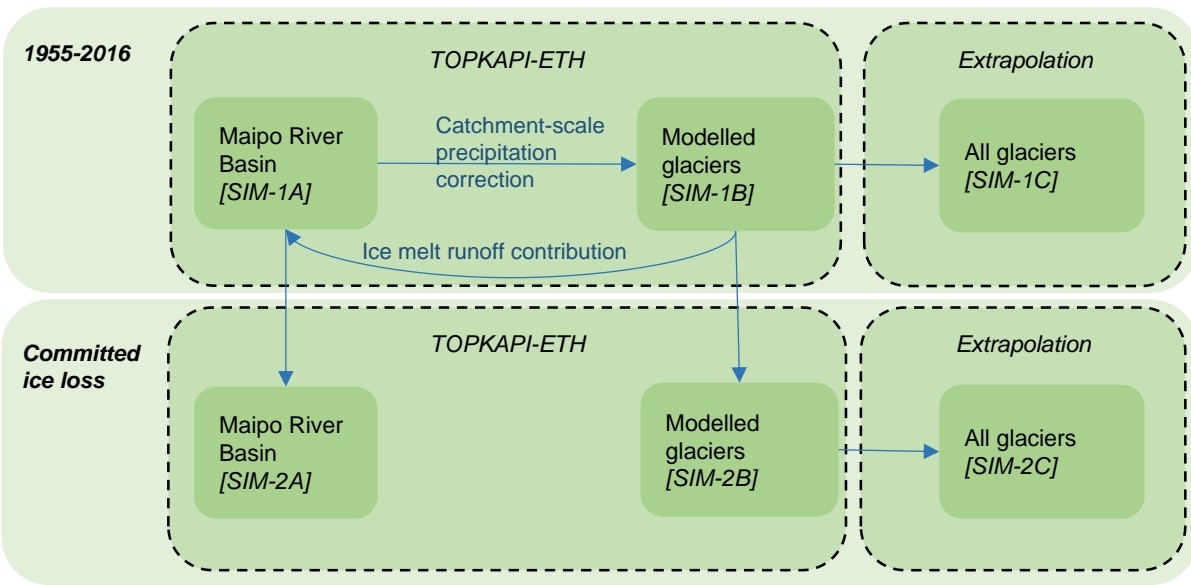

**Figure 5: Organization of the simulations. The two main boxes indicate the time period of the simulations (1955-2016 and committed ice loss scenario), the boxes with a dashed outline indicate the method used (TOPKAPI-ETH or extrapolation), and the smallest boxes indicate the spatial domain (Maipo River Basin, modelled glaciers, and all glaciers). The arrows indicate outputs that are used in other methods or domains. The codes in brackets (e.g. SIM-1A) correspond to the simulation codes defined in Table S1.**



**Figure 6: Variability of meteorological and glaciological variables in the Maipo River Basin over the period 1955-2016. (a) Air temperature and precipitation with a 3-year moving mean, (b) equilibrium line altitude (ELA), (c) cumulative glacier mass balance for the modelled glaciers (simulated with TOPKAPI-ETH), the entire basin (extrapolation), and the measurements on Echaurren Norte Glacier, and (d) cumulative glacier mass balance for each sub-catchment. In b), the difference between the ELA in the last 10 years (2006-2016) and the first 10 years (1955-1965) of the study periods is indicated, as well as the equivalent ELA increase rate. The shadowed area in (b) shows the standard deviation of the elevation of grid cells with a mass balance between –0.1 m w.e and 0.1 m w.e.**





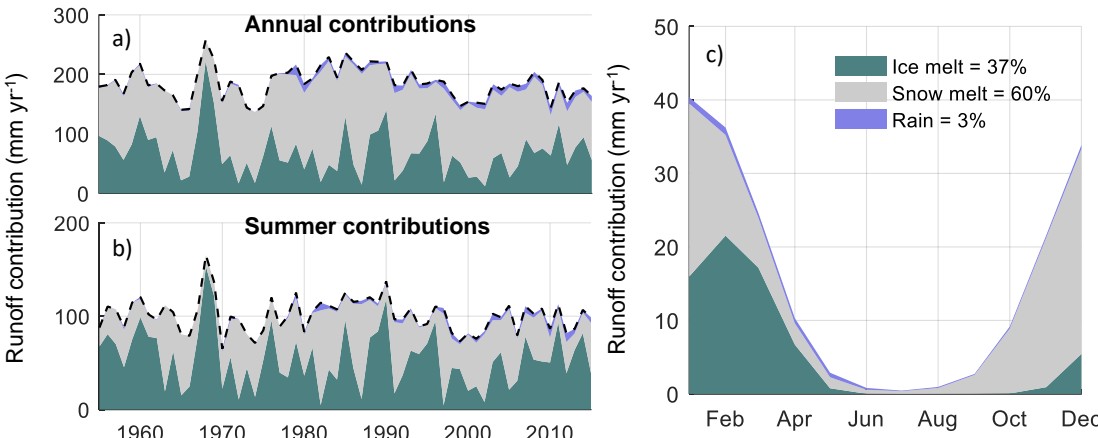

**Figure 7: Runoff contribution from ice melt, snow melt and rain from the headwater regions defined by the 1955 glacierized areas. The units are normalized by the Maipo River Basin area. (a) Total annual contribution, (b) summer contribution, and (c) seasonal average contribution. The percentage of each contribution over the period 1955-2016 are indicated next to the legend.**



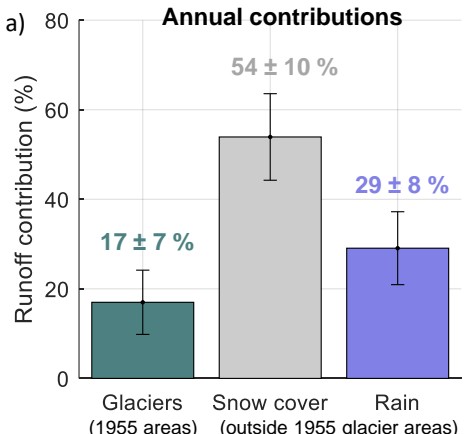
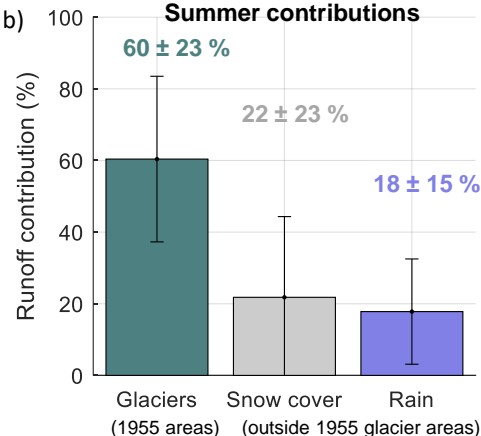

**Figure 8: Partition of runoff contribution in the Maipo River Basin in the period 1955-2000. The contributions are computed for the headwater regions defined by the 1955 glacierized areas (the sum of ice melt, snow melt and rain), and snow melt and rain outside those areas. The plus/minus symbol refers to the inter-annual variability. (a) Annual, and (b) summer contribution.**





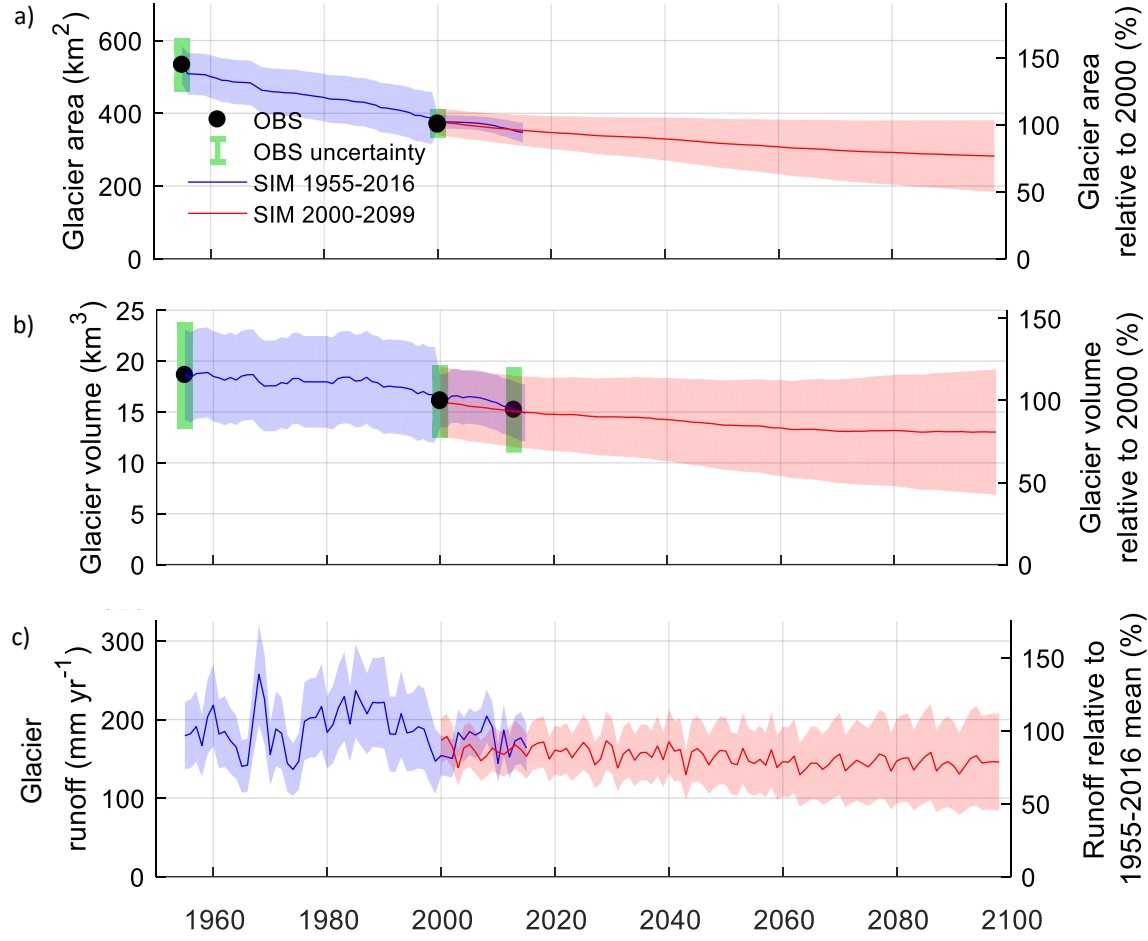

**Figure 9: Variations in (a) ice volume, (b) glacierized areas, and (c) glacier runoff in the Maipo River Basin for the past period**
**(1955-2016) and the committed ice loss scenarios assuming a constant climate. In a) and b) we use results from the glacier inventories, and the combination of ice thickness estimates and geodetic mass balances, respectively, as observations of glacier area and volume. Glacier runoff in (c) is computed for the Maipo River Basin (i.e. runoff units are normalized by the basin area). While the uncertainty bars of the observations are shown in green, that of the simulations are shown in blue for the 1955-2016 period and in red for the committed ice loss scenarios. The committed ice scenarios do not represent a realistic projection for the future, and**
**we use the years of 2000 to 2100 in the x-axis for visual purposes only.**

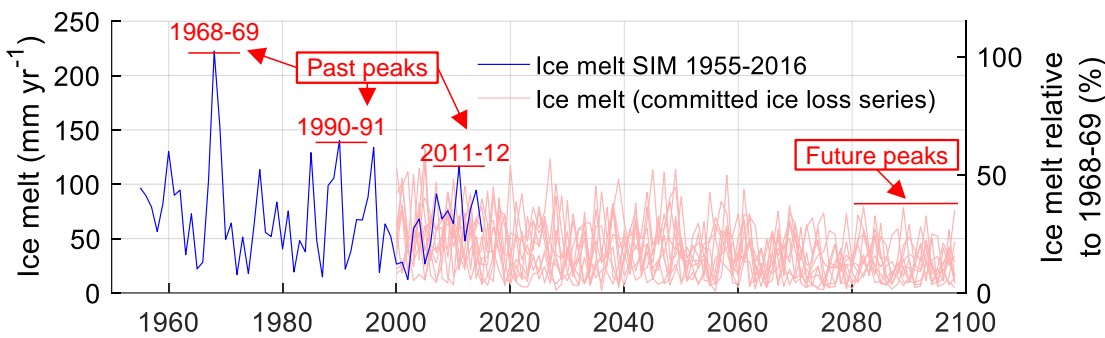

**Figure 10: Variations in ice melt in the Maipo River Basin for the past period (1955-2016, in blue) and each one of the ten committed ice loss scenarios (light red). The peaks of ice melt over the past period and those at the final decade of the committed ice loss scenarios are highlighted in red. On the right axis, we set the ice melt estimated for the severe drought of 1968-1969 as 100%. Ice melt is computed for the Maipo River Basin (i.e. runoff units are normalized by the basin area). The committed ice scenarios do not represent a realistic projection for the future, and we use the years of 2000 to 2100 in the x-axis for visual purposes only.**





**Table 1: Parameters in TOPKAPI-ETH's snow and ice modules for the 1955-2016 time period**

| Module | Parameter | Symbol | Calibrated value Simulation Individual glaciers | Maipo River Basin | Units |
|---|---|---|---|---|---|
| Snow accumulation and gravitational transport | Snow/rain threshold | $P_T$ | 0 | 2 | °C |
| | Snow holding capacity parameter 1 | $SRF_c$ | 250 | 250 | m |
| | Snow holding capacity parameter 2 | $SRF_a$ | 0.172 | 0.172 | - |
| ETI model | Shortwave radiation factor | $SRF$ | 0.002-0.0140 | 0.0090 | mm m$^2$ h$^{-1}$ W$^{-2}$ |
| | Air temperature factor | $TF$ | 0-0.4 | 0.01 | mm h$^{-1}$ °C |
| | Air temperature threshold for the onset of melt | $T_T$ | 0 | 1 | °C |
| Sub-debris ice melt | Shortwave radiation factor | $SRF_d$ | 0.25*SRF | - | mm m$^2$ h$^{-1}$ W$^{-2}$ |
| | Air temperature factor | $TF_d$ | 0.25*TF | - | mm h$^{-1}$ °C |
| | Albedo debris | $\alpha_{debris}$ | 0.16 | - | |
| Surface albedo | Albedo of fresh snow | $\alpha_1$ | 0.83 | 0.90 | |
| | Decay of snow albedo | $\alpha_2$ | 0.11 | 0.11 | |
| | Ice albedo | $\alpha_{ice}$ | 0.3 | - | |





**Table 2: Morphological properties of the 26 glaciers modelled with TOPKAPI-ETH**

| Property | Range for modelled glaciers | Total range |
|---|---|---|
| Area (km$^2$) | 1.1 – 21.3 | 0.01 – 21.3 |
| Mean elevation (m a.s.l.) | 3313 – 4526 | 2801 – 6174 |
| Slope (°) | 10.2 – 26.6 | 6.3 – 60.7 |
| Aspect (southing) (°) | 90.4 – 178.9 | 1.1 – 179.6 |
| Debris coverage (%) | 0 – 95 | 0 – 100 |




**Table 3: Results of the multiple regression analysis for the committed ice loss scenario simulations**

| Property | Fraction of explained variance of the model (%) | Sign of the mass balance dependence |
|---|---|---|
| Area (km²) | 42.1 | − |
| Median elevation (m a.s.l.) | 18.2 | + |
| Percentage of debris cover (%) | 17.1 | + |
| Sky view factor (%) | 15.6 | + |
| Aspect (southing) (°) | 7.0 | + |