# Peer review of "Glacier runoff variations since 1955 in the Maipo River Basin, semiarid Andes of central Chile"

_The Cryosphere, 2019_

## Referee Comment (RC1) · Francisca Bown (Referee) · 9 Jan 2020

GENERAL COMMENTS

The study examines the glacier mass balances for the upper Maipo, central Chile, between years 1955 and 2000/13 and the corresponding melting water contributions to runoff over that period. This is done by physically-based modelling of selected glaciers and its extrapolation to the entire basin. The approach has been tested abroad and now adapted for the Andes setting for a period that concurs the largest observed retreat rates in historical times.

Input glaciological data are two main glacier inventories separated by 48 years, originated from very different type of sources, resolution, precision, etc, but properly cor-

rected and processed for the purposes of direct comparison as best as possible. These were complemented to Digital Elevation Models (DEMs) of same dates, distributed ice thicknesses obtained from modelling & geodetical balances, and several types of hydro-meteorological datasets (mostly downscaling reanalysis and remotely-sensed data i.e. input local observations are limited). Extrapolations (spatially and temporarily), calibrations and verifications are careful. It is clear, however, that lack of direct radar measurements and AWS data over glaciers must have committed the results at some extent. This is particularly true when authors raised datasets discrepancies and provide sublimation estimates without in situ verification. In that sense, TOPKAKI-ETH would require more field measurements than applied for an optimal hydrological simulation.

Ice volume and runoff values and trends are given in reasonable orders of magnitude and complement former studies in the region. The authors raised that typical increasing or decreasing phases of peak water cannot be observed over the period 1955-2016, however there is a bulk of facts (i.e. areal and ice volume losses, negative mass balances and elevation changes, observed runoff trends and conservative committed ice losses up to year 2100) that suggests this peak is hidden somewhere within 2000-10. In contrast, authors argue a possible transient equilibrium with climate of some glaciers to justify some short periods of positive/neutral mass balances, hyphotesis which is not really supported.

Apart from that, the study is clearly explained from beginning to end, it is a well-structured & written manuscript. Figures, tables and supplementaries are generally all informative and of appropriate visual quality, but with some improvements and clarifications I would recommend. I particularly missed a table providing mass balance and runoff values per each sub-basin, which would make more explicit and/or highlight possible influence of factors such as elevation range and latitude.

The study settled the hydrological role of glaciers together with those of snow and rain, both on annual and seasonal basis. This is helpful in current times when concerns

on water security are quite high and general public receives distorted information from environmental NGOs. It additionally provides the main forcing factors of hydrological trends and predicts the decreasing glacier buffer capacity even at the conservative scenario. By themselves, these points suggest an important impact in the scientific community, likely for stakeholders and decision makers as well.

There are much more strengths than weaknesses that make this manuscript suitable for going from TCD into TC after very minor editing.

SPECIFIC COMMENTS

Lines 23-26: "If glaciers in the basin were in equilibrium with the climate of the last two decades, their volume would be reduced to 81±38% of the year 2000 volume, and glacier runoff during dry periods would be 61±24% of its maximum contribution in the period 1955-2016, considerably decreasing the drought mitigation capacity of the basin". This sentence refers an optimistic scenario based on minimum ice volumetric loss and minimum decrease of glacier runoff contribution, but it is rather confusing and probably needs improvement in redaction, probably in a way like this or similar: "Assuming conservative ice losses of 81% under a constant climate…glacier runoff during dry periods…"

Lines 83-84: "Unrealistic" mentioned several times seems awkward.

Lines 127-128: Inventories error assignments of 5 (year 2000) and 10% (year 1955) seem rather arbitrary. Can you explain better?

Lines 161-173: When calculating ice thicknesses in 1955 based on Huss and Farinotti complemented to geodetic balances 1955-2000 and area-volume ratio, there is an intrinsic assumption of no basal melting. I think this could be mentioned.

Lines 176-177: Uncertainty of 15% in average for 1955, 2000 and 2013? 1955 is clearly more uncertain, maybe you could clarify.

Lines 179-203: Is there any particular reason why fluviometric data elsewhere available

upstream El Manzano was not used for feeding or verifying the model results?

Lines 204-212: Modis datasets used in calibration of snow processes have minor resolution than the model output. Something to say about that?

Lines 237-240: "To calculate ice melt under supra-glacial debris we also use the ETI model but with reduced melt factors (see section 4.1.3). Although TOPKAPI-ETH includes a melt module..." I understand it, but be aware there is a bias. Debris impact on melt can be variable depending on thickness, mineralogy, etc.

Lines 282-287: Because of different conditions of elevation ranges, air humidity, winds, etc, among 5 sub-basins, I disagree with the representativeness of 34 mm/yr of sublimation, at least in the case of the higher ones. I think authors should raise there is a limitation of SWE information from Landsat.

Lines 288: "...and is in the order of the model uncertainties (see Figure 2)." You mean 34 mm/yr in comparison to 49.9 mm of RMSE? Please clarify.

Lines 317-319: "We suspect that this is an expression of the fact that some of the processes not included in TOPKAPI-ETH (namely permafrost, sublimation, snow dynamics or geothermal fluxes) may play a role governing the mass balance of these glaciers". Then it is partially contradictory to this sentence: "...which is a reasonable estimate of sublimation amounts for this region...".

Lines 333-335: "Interestingly, several of the glaciers show a positive or near-neutral mass balance over the entire period, which might be an indication that these glaciers have already retreated close to a new equilibrium." This seems to contradict evidence of glacier mass balances in the entire Andes.

Lines 410: Authors report an important and larger ELA elevation than reported in Carrasco et al (2005). It should be highlighted.

Lines 419-410: "In general, glaciers in southern catchments show more positive mass balance than those in northern catchments." This occurs despite elevations are much

lower. Any explanation other than precipitation?

Lines 424-440: This is the core of this research. It compares the contributions of ice, snow and rain in annual and summer basis. Is the 3% decrease of glacier summer contribution (entire study period versus current drought) a possible indication peak water was already reached?

Lines 441-455 & Figure 9c: Maybe a "realistic" projection could have complemented this analysis.

Lines 481-485: As raised by the authors, difference in mass balances among sub-basins can depend on many climatic and morphological factors, however it is doubtful that precipitation increases that much in semiarid Andes to lead positive mass balances in southern basins. Unless there is data enough to support this statement.

TECHNICAL CORRECTIONS

Line 164: " a meaningful 1955 ice thicknesess…" Delete "a"

Lines 256-261: I think this sentence repeats information from section 3.3.

Lines 514-542 Uncertainties of modelling I particularly find this could have been assessed in summary at the end of methods section.

Figure 1 (a): Maipo outline may be better recognised if Chile and Argentina are just outlined (without color filling); (b) debris-free areas could be coloured in blue because white is difficult to distinguish over yellow; (c) I would recommend sub-basins labels to be horizontally oriented with brackets, so far I cannot tell where are the boundaries between them; (d) Why Volcán label and number of glaciers are in light grey?

Legend Figure 1: "a) Maipo River Basin next to the city of Santiago, in central Chile; (b) the basin outlet and the sub-catchments, rivers, main glaciers, and hydro-meteorological stations…" These are not all glaciers, nor the 26 modelled glaciers, just main ones.

Figure 2 (a): It is Cortes et al 2016 or Cortes and Margulis 2017? Please clarify.

Figure 3. I am not sure if this is necessary (instead they could be shown in Figure 1(a). In any case, glaciers in white are difficult to distinguish over yellow. Maybe blue is more appropriate. Name of main glaciers could be added.

---

## Referee Comment (RC2) · Anonymous Referee #2 · 14 Feb 2020

Review of "Glacier runoff variations since 1955 in the Maipo River Basin, semiarid Andes of central Chile" Authors: Álvaro Ayala, David Farías-Barahona, Matthias Huss, Francesca Pellicciotti, James McPhee, Daniel Farinotti. This article aims to quantify the evolution of the glaciers and the runoff since 1955 in an Andean Chilean catchment using the TOPKAPI-ETH hydro-glaciological model. This study is very interesting and could help for water management in this region. Nevertheless, some issues have to be resolved before publication in the TC journal (see below).

1 – I am not convinced by the long term simulations that use a 'stationary climate' during two decades. By nature, climate is non stationary and an important decadal variability is observed in this region for the different climate variables (for instance the precipitation). The simulations provided here can be a first approach but simulations

[Figure]

based on future climate projections should be made. This is important if one consider that this kind of study is oriented to water ressource management (as state in various places in the article). 2 – The methodology to calculate volume and surface glacier variations in relation with the climate is confused. More details should be done (time step, kind of processes, basal sliding, etc. . . .). 3 – Concerning the precipitation used in the model, a clear explanation on how the discrimitation phase between solid and liquid is done is missing. I don't understand why an additional meteorological station is used here. Only one station is not adequate to the size of the catchment. A correction is made on the raw precipitation but details should be given concernig the methodology used. Finally how does the model compute the sublimation ? 4 – For the snow cover evolution, no in-situ data is provided. Does such data exists ? If yes a comparison best-ween the simulations of the snow cover with TOPKAPI and CAMEL-CL models should be made. Please give more details concerning the CAMEL-CL product (resolution, etc. . .). 5 – Recent studies underlined the importance of groundwater in mountainous catchments. Here in the model, it seems that no water flux into the ground exists. This can not be true. Subteranean water fluxes may have an importance for the future of water ressources. 6 - Please define 'glacier runoff'. 7 - If the model is oriented to be used for water management (as stated in the lines 544-547), please give results for daily simulations. What is the agreement between the simulations and the observations at daily time-step ? 8 - I think that all the sections 6.3, should be moved at the beginning of the result section.

Specific comments : Abstract – line 14 : please precise the time step of the simulated runoff Abstract – line 20 : please precise the latitude range Abstract – Please precise if the glacier area's changes are taking into account in the model Line 81 : Ấń . . ..estimate glacier changes. . .. Ấż please precise if it is surface, volume or both ? Line 95 : Ấń . . .., to which it provides most of the drinking water. . . Ấż please specifiy the percentage (give a quantity). Line 125 : If I understand well, the outlines taken in 1955 are used for the year 2000 ? If the answer is yes, it is certainly not true. Please add more details. Lines 191-194 : Ấń . . ..for the years 2004-2016 . . ... Ấż How do you do before 2004 ?

Line 221 : The model is Âń physically-oriented Âż, so how do you do with the land cover and land use changes over the last decades ? Please indicate clearly that in the article. If the changes are important, this should be taken into account in the model. Line 245 : Âń . . ..but no area increases due to positive mass balance are prescribed. Âż This is not a valid statement as it is possible to observe glacier's advances. So if the model is Âń physically-oriented Âż this should be changed. In all this part, the time-step should be precised. In the model, the selected calibration and validation periods are unclear. Furthermore, details should be given concerning the snwofall/rainfall discrimitation ($T^\circ$ threshold ?). I don't understand why the ERA-interim and MERRA products are not tested in the model. Please explain why. Line 315 : What is the criterion Âń to fit the geodetic mass balances. . .. Âż ? Line 366 – 367 : Please rewrite this sentence, unclear. Fig. 1 : please specify in the legend how the total ice volume is obtained Fig. 6 : you should add the uncertainty for each curve. Fig. 7 : Where is the subteranean part ? Please indicate the evolution of glacier volume and glacier area. Fig. 8 : In the figure 7, you indicate Rain = 3% and in the figure 8 you indicate Rain : 29+/-8 % , why ? Tab. 1 : Please indicate the tested ranges for each parameter and the references associated.
* * *

---

## Author Comment (AC2) · 28 Mar 2020

The comment was uploaded in the form of a supplement:
https://www.the-cryosphere-discuss.net/tc-2019-233/tc-2019-233-AC2-supplement.pdf
* * *

---

## Author Response (AR1)

**Glacier runoff variations since 1955 in the Maipo River Basin, semiarid Andes of central Chile**

**Response to reviewers**

Reviewers: Black font

Authors: Blue font

We thank both reviewers for evaluating our article, for highlighting the relevance of our study, as well as for their useful feedback and comments. We have responded to all the questions raised by the reviewers, and provided a detailed justification where we did not perform the suggested changes.

We summarize the main changes in our manuscript below:

1. As suggested by reviewer 1, we have added a new table (Table 4) that summarizes glacier mass balance and runoff contribution per sub-catchment in the historical period (1955-2016), and in the period when glaciers have reached an equilibrium with the current climate (last 20 years of the committed ice loss scenarios).

2. As the two reviewers pointed out that the model was validated against few field data, we have added a new sub-section (S3) called "Additional model validation" to the supplementary material. The sub-section compares model results against i) streamflow records and intermediate sections of the Maipo River and ii) a 40-year time series of SWE manual measurements. We found that our simulations compare well with these point-scale observations. This new section has thus strengthened our article.

3. We found that some of the scripts that analysed the results from TOPKAPI-ETH and the extrapolation methodology used a number of 558 km$^2$ for total glacierized area in 1955, instead of 532 km$^2$, which is the correct number. The number of 558 km$^2$ was also used twice in the text. After correcting the scripts, some numbers changed in the text and in some plots (figures 2, 7, 8, 9c and 10), but all the changes are small and do not affect our conclusions.

4. We have improved figures 1, 3 and 6, following suggestions from both reviewers.

5. A number of small changes have been performed in the text, following suggestions from both reviewers.

Please see our detailed responses below.

**Response to reviewer 1 (Francisca Bown)**

The study examines the glacier mass balances for the upper Maipo, central Chile, between years 1955 and 2000/13 and the corresponding melting water contributions to runoff over that period. This is done by physically-based modelling of selected glaciers and its extrapolation to the entire basin. The approach has been tested abroad and now adapted for the Andes setting for a period that concurs the largest observed retreat rates in historical times. Input glaciological data are two main glacier inventories separated by 48 years, originated from very different type of sources, resolution, precision, etc, but properly corrected and processed for the purposes of direct comparison as best as possible. These were complemented to Digital Elevation Models (DEMs) of same dates, distributed ice thicknesses obtained from modelling & geodetical balances, and several types of hydro-meteorological datasets (mostly downscaling reanalysis and remotely-sensed data i.e. input local observations are limited). Extrapolations (spatially and temporarily), calibrations and verifications are careful.

We thank the reviewer for her thorough evaluation of our article, and for her useful suggestions and comments. We have responded to all the questions raised by the reviewer. Please see our detailed responses below.

It is clear, however, that lack of direct radar measurements and AWS data over glaciers must have committed the results at some extent. This is particularly true when authors raised datasets discrepancies and provide sublimation estimates without in situ verification. In that sense, TOPKAKI-ETH would require more field measurements than applied for an optimal hydrological simulation.

We very much agree with the reviewer that more field data would be useful and they could reduce the uncertainty in our results. In particular, more field data can be useful to better constrain our estimates of precipitation and temperature at remote high-elevation areas (such as Tupungatito Volcano or the Upper Maipo sub-catchment), and improve the simulation of some specific processes, such as elevation changes due to ice flow, the impact of supraglacial debris on glacier melt, or long-term ice albedo changes.

The reason why we did not use more field data however is that these are not widely available in the region. However, to alleviate the difficulties posed by the lack of a basin-wide set of data described in the previous paragraph, we have made a consistent effort to derive most of the Topkapi-ETH parameters from data collected in previous field campaigns in this region, starting in 2008. The datasets collected in those campaigns consisted of ablation stakes, distributed snow depth measurements, on-glacier meteorological records, terrestrial cameras, and others, that we have used in previous studies to force models of variable complexity that we now fully exploit in this study (Pellicciotti et al., 2008; Ragettli and Pellicciotti, 2012; Ayala et al., 2016, 2017a, 2017b; Burger et al., 2019). In particular, we use the previous, field-based modelling to calibrate several TOPKAPI-ETH parameters, such as melt factors, albedo decay rates, and parameters controlling the snow gravitational distribution. These previous studies have also shown that many of the parameters required by the model are fairly stable, in the sense that they can be extrapolated from one glacierized area to another with a reasonable degree of confidence (Ayala et al., 2017b; Burger et al., 2019). We thus indirectly use a relatively (for the region) large amount of field data to inform the model and calibrate its parameters. In relation to sublimation, Corripio (2003) and Ayala et al. (2017a, 2017b) estimated its daily and seasonal rates at several sites across the semiarid Andes, and we use these estimates as a reference in our study.

In the revised version, we include these ideas in section 6.3 (lines 578-586):

"Although our study has benefited from a series of new meteorological and glaciological datasets presented for the Southern Andes in recent years (Cortés and Margulis, 2017; Álvarez-Garretón et al., 2018; Farías-Barahona et al., 2019a), the lack of field data in the Maipo River Basin is something that needs to be taken into account in glacio-hydrological modelling studies in the region, particularly at high-elevation, remote sites. In this study, we alleviate the difficulties posed by the lack of basin-wide field data, and its impact on the TOPKAPI-ETH results, by deriving most of the model parameters from data collected in previous field campaigns in this region, starting in 2008 (Pellicciotti et al., 2008; Ragettli and Pellicciotti, 2012; Ayala et al., 2016). These previous studies have also shown that many of the parameters required by the model are fairly stable, in the sense that they can be extrapolated from one glacierized area to another with a reasonable degree of confidence (Ragettli et al., 2014; Ayala et al., 2017b; Burger et al., 2019)."

In relation to the two specific datasets mentioned by the reviewer (radar data and on-glacier AWSs), we note that radar data from the Glacier Thickness Database (GlaThiDa) (Gärtner-Roer et al., 2014) are used as validation for our ice thickness estimates for Volcán Tupungatito and Marmolejo glaciers (lines 162-165). We also notice, with respect to the reviewer's appropriate comment on lack of on-glacier AWSs, that a glacio-hydrological model such as TOPKAPI-ETH applied at the large scale of the entire Maipo basin cannot be forced with on-glacier data, as these represent the atmosphere in the glacier boundary layer and would result in incorrect estimates of all the remaining hydrological components in the non-glacierised sections of the catchment.

Ice volume and runoff values and trends are given in reasonable orders of magnitude and complement former studies in the region. The authors raised that typical increasing or decreasing phases of peak water cannot be observed over the period 1955-2016, however there is a bulk of facts (i.e. areal and ice volume losses, negative mass balances and elevation changes, observed runoff trends and conservative committed ice losses up to year 2100) that suggests this peak is hidden somewhere within 2000-10.

Based on our data and results, we think that there is not enough evidence to identify a clear peak water in our study period. Since glacier runoff is defined as the summed contributions of rain, snowmelt and ice melt over the areas defined by the glacierized areas in 1955, peak water is not only connected to ice melt, but also to the annual variability of precipitation. Given the humid years in the 1980s and the large values of ice melt in the 1990s, we believe that glacier runoff in the 2000s was actually lower than that in the previous decades (see Figure 9c). In addition, we think that the exact occurrence of peak water will depend also on future changes (e.g. more precipitation or more ice melt), which are not addressed in our article.

These arguments have been summarized in our conclusion "b" (lines 617-619) as:

"Instead of a clear peak water, we identify a decreasing sequence of runoff maxima that can be linked to both a decrease in precipitation since the 1980s and a reduction of ice melt. The exact occurrence of peak water will depend also on future changes (e.g. more precipitation or more ice melt), which are not addressed in our article"

In contrast, authors argue a possible transient equilibrium with climate of some glaciers to justify some short periods of positive/neutral mass balances, hyphotesis which is not really supported.

To our knowledge, short periods of positive/neutral mass balance in this region are well documented in the literature. Although glaciers in the semiarid Andes have been retreating for several decades, the direct mass balance

measurements at Echaurren Norte Glacier, and the latest geodetic mass balance studies point to near-neutral glacier mass budgets in that decade. We have summarised this evidence in the table below.

**Summary of glacier mass balance results in the Central Andes for the 2000-2013 period**

| Domain | Type of data | Value | Period | Source |
|---|---|---|---|---|
| Echaurren Norte Glacier | Glaciological mass balance | +0.2 m w.e. yr$^{-1}$ | 2000-2009 | World Glacier Monitoring Centre (WGMS) |
| Echaurren Norte Glacier | Geodetic mass balance | +0.54±0.40 m w.e. yr$^{-1}$ | 2000-2009 | Farías-Barahona et al.(2019) |
| Central Andes | Geodetic mass balance | +0.17±0.23 m w.e. yr$^{-1}$ | 2000-2009 | Dussaillant et al. (2019) |
| Bello Glacier | Model simulations | –0.01±0.09 m yr$^{-1}$ | 2000-2013 | Burger et al. (2019) |
| Yeso Glacier | | –0.03±0.09 m yr$^{-1}$ | | |

Apart from that, the study is clearly explained from beginning to end, it is a well- structured & written manuscript. Figures, tables and supplementaries are generally all informative and of appropriate visual quality, but with some improvements and clarifications I would recommend. I particularly missed a table providing mass balance and runoff values per each sub-basin, which would make more explicit and/or highlight possible influence of factors such as elevation range and latitude.

This is a very useful suggestion and we include the suggested table in the revised version of the article (Table 4):

**Table 4: Simulated glacier mass balance and runoff in the sub-catchments compared with their main characteristics**

| Basin | Mean elev. (m a.s.l.) | Mean lat. (°S) | Glacie rized area in 1955 (km$^2$) | Average annual glacier mass balance in 1955-2016 (m w.e. yr$^{-1}$) | Runoff contribution in 1955-2016 (*) (mm w.e. yr$^{-1}$) | | Runoff contribution in the committed ice loss scenarios (mm w.e. yr$^{-1}$) | |
|---|---|---|---|---|---|---|---|---|
| | | | | | Total | Ice melt | Total | Ice melt |
| Olivares | 3698 | 33.3 | 111 | –0.26 ± 0.07 | 34.1 ± 7.9 | 15.8 ± 3.6 | 22.5 ± 6.1 | 5.4 ± 1.5 |
| Colorado | 3755 | 33.4 | 152 | –0.10 ± 0.07 | 53.2 ± 12.2 | 16.1 ± 3.7 | 42.7 ± 11.5 | 6.4 ± 1.7 |
| Yeso | 3303 | 33.7 | 65 | –0.09 ± 0.07 | 21.5 ± 4.9 | 7.5 ± 1.7 | 17.1 ± 4.6 | 3.6 ± 1.0 |
| Volcán | 3392 | 33.8 | 86 | +0.04 ± 0.07 | 24.2 ± 5.6 | 7.7 ± 1.8 | 20.0 ± 5.4 | 3.5 ± 1.0 |
| Upper Maipo | 3182 | 34.0 | 111 | –0.03 ± 0.07 | 41.8 ± 9.6 | 12.6 ± 2.9 | 33.4 ± 9.0 | 4.4 ± 1.2 |

| Maipo River Basin | 3175 | 33.6 | 532 | −0.09 ± 0.07 | 176.9 ± 40.7 | 65.5 ± 15.1 | 138.6 ± 37.4 | 25.8 ± 7.0 |

(*) From the areas defined by the 1955 glacier outlines, but normalized by the Maipo River Basin area

The study settled the hydrological role of glaciers together with those of snow and rain, both on annual and seasonal basis. This is helpful in current times when concerns on water security are quite high and general public receives distorted information from environmental NGOs. It additionally provides the main forcing factors of hydrological trends and predicts the decreasing glacier buffer capacity even at the conservative scenario. By themselves, these points suggest an important impact in the scientific community, likely for stakeholders and decision makers as well. There are much more strengths than weaknesses that make this manuscript suitable for going from TCD into TC after very minor editing.

We thank again the reviewer for her positive comments about our article. Please see our responses to the specific comments and technical corrections below.

**SPECIFIC COMMENTS**

Lines 23-26: "If glaciers in the basin were in equilibrium with the climate of the last two decades, their volume would be reduced to 81±38% of the year 2000 volume, and glacier runoff during dry periods would be 61±24% of its maximum contribution inthe period 1955-2016, considerably decreasing the drought mitigation capacity of the basin". This sentence refers an optimistic scenario based on minimum ice volumetric loss and minimum decrease of glacier runoff contribution, but it is rather confusing and probably needs improvement in redaction, probably in a way like this or similar: "Assuming conservative ice losses of 81% under a constant climate...glacier runoff during dry periods..."

We apologise if the wording was confusing. In fact, we are neither assuming any ice loss (as in the sentence suggested by the reviewer) nor is the value of 81% resulting from assuming a constant climate. We have changed these sentences to (lines 24-26):

"Glaciers in the Maipo River Basin will continue retreating because they are not in equilibrium with the current climate. In a hypothetical constant climate scenario, glacier volume would reduce to 81±38% of the year 2000 volume, and glacier runoff would be 78±30% of the 1955-2016 average. This would considerably decrease the drought mitigation capacity of the basin."

Lines 83-84: "Unrealistic" mentioned several times seems awkward.

To avoid word repetition, we have reworded some of the sentences that included the idea of "unrealistic projections".

- (lines 536-537): "We stress that these estimates do not correspond to a realistic future scenario, but are an indication of the glacier changes that past climate will produce in any case." is changed to "We stress that these estimates are an indication of the glacier changes that past climate will produce in any case."
- Captions of figures 9 and 10: "The committed ice loss scenarios do not represent a realistic projection for the future, and we use the years of 2000 to 2100 in the x-axis for visual purposes only" is changed to "For visual purposes, we present the committed ice loss scenarios using the period 2000-2100 in the x-axis."

Lines 127-128: Inventories error assignments of 5 (year 2000) and 10% (year 1955) seem rather arbitrary. Can you explain better?

We have now extended the explanations using these sentences (section 3.1, lines 130-133):

"In this study, we assign an error of 5% to the year 2000 inventory, which is a common choice for glacier inventories (Paul et al., 2013), and has been used for this inventory in particular (Barcaza et al., 2017). As the inventory of 1955 suffers from additional errors (such as the presence of snow patches that likely made the interpretation of glacierets difficult, and the use of Lliboutry maps to fill missing areas), we assume an error of 10% for that year."

Lines 161-173: When calculating ice thicknesses in 1955 based on Huss and Farinotti complemented to geodetic balances 1955-2000 and area-volume ratio, there is an intrinsic assumption of no basal melting. I think this could be mentioned.

We agree with the reviewer. We have now added the following sentence (section 3.2, lines 184-186):

"In the calculation of glacier volumes, we implicitly assume that no basal melting takes place. The error introduced by neglecting this process is much less than the uncertainty associated with the ice thickness estimates and the geodetic mass balance."

Lines 176-177: Uncertainty of 15% in average for 1955, 2000 and 2013? 1955 is clearly more uncertain, maybe you could clarify.

Thanks for noting this. The uncertainty of 15% in total ice volume is only for 2000. The uncertainty in the total ice volume in 1955 and 2013 is larger because it includes the uncertainty from the geodetic mass balance. We have changed this to (lines 181-186):

"For the total ice volume of the investigated basin, we assume an uncertainty of 15% in year 2000. (…) The uncertainty in the total ice volume in 1955 and 2013 is larger than in 2000 since it also includes the uncertainty from the geodetic mass balances."

Lines 179-203: Is there any particular reason why fluviometric data elsewhere available upstream El Manzano was not used for feeding or verifying the model results?

This is a very useful comment. We used only the Maipo en El Manzano streamflow gauge because we focused on the glacier runoff contributions at the scale of the entire catchment. To extend the verification of the model as suggested by the reviewer, a new sub-section ("Additional model validation") is included in the supplementary information of the revised version. In this sub-section, we add new figures and tables showing the verification of the model at i) six intermediate streamflow gauges, and ii) the Laguna Negra snow monitoring site, at which the Chilean Directory of Water Resources (DGA) have measured annual near-maximum SWE for several decades. Due to its length (5 pages), this new sub-section is included at the end of this document.

Lines 204-212: Modis datasets used in calibration of snow processes have minor resolution than the model output. Something to say about that?

The MODIS datasets are used only for the calibration of snow parameters in the basin-wide model, which has a spatial resolution of 1 km. The 1-km model resolution is actually lower than that of the MODIS datasets (500 m) and that of the SWE reconstruction (180 m). In any case, all datasets and results are aggregated at the catchment

scale and we used basin-wide values in the model calibration. We think that given the large size of the Maipo River Basin, the differences in spatial resolution will not strongly affect the parameter calibration.

We have now provided more details (section 4.1.2, lines 282-284):

"The calibration of the Maipo River Basin model was performed for the period April 2003 to March 2016, and consists of two steps: (i) the snow parameters are varied in order to fit SCA and SWE aggregated at the scale of the entire basin from the MODIS and CAMELS-CL datasets (section 3.4), …"

Lines 237-240: "To calculate ice melt under supra-glacial debris we also use the ETI model but with reduced melt factors (see section 4.1.3). Although TOPKAPI-ETH includes a melt module..." I understand it, but be aware there is a bias. Debris impact on melt can be variable depending on thickness, mineralogy, etc.

To account for this comment, we have added the next sentence (lines 258-260):

"As a result of our assumptions, we expect that some of the spatial patterns of glacier ablation induced by the spatial variability of supraglacial debris thickness are not accurately represented in our simulations."

Lines 282-287: Because of different conditions of elevation ranges, air humidity, winds, etc, among 5 sub-basins, I disagree with the representativeness of 34 mm/yr of sublimation, at least in the case of the higher ones.

We agree with the reviewer about the large spatial variability of surface sublimation. Please note that the 34 mm $yr^{-1}$ correspond only to a basin average of discarded snow and that we do not assume that the value applies to all five sub-basins individually. In general, we estimate about 688 mm yr-1 (~2 mm $day^{-1}$) in the areas above 5000 m a.s.l., which is in agreement with estimates derived from energy balance models in the region (Corripio, 2003; Ayala et al., 2017a, 2017b).

To be more precise about this topic, we have added the following sentences:

(Section 4.1.2, lines 313-314) "As elevation decreases south, the discarded snow varies from about 121 mm w.e. $yr^{-1}$ over the Colorado sub-catchment to about 10 mm w.e. $yr^{-1}$ over Upper Maipo."

(Section 4.1.3, lines 341-343) "However, as these models are calibrated on volume loss (thus including both losses by sublimation and melting), it can be assumed that glacier response is well captured, but the portioning of hydrological fluxes (sublimation versus runoff) is unconstrained."

I think authors should raise there is a limitation of SWE information from Landsat.

We think that we have not been clear enough in our text. Please consider that SWE does not come directly from Landsat but from inversion of snowmelt calculations from re-analysis using Landsat images as boundaries. We have now included the following sentence (section 3.4). Note that the sentence also addresses the limitation of the Landsat based SWE product (lines 219-225):

"The SWE reconstruction was obtained from a data assimilation framework that integrates a land surface and depletion model, the assimilation of Landsat imagery, and the Modern Era Retrospective Analysis for Research and Applications (MERRA) reanalysis as a forcing dataset (Cortés et al., 2016; Cortés and Margulis, 2017). Although not all physical processes are included in the assimilation process (for example, blowing snow sublimation), the dataset has been validated at several sites across the Southern Andes (Cortés et al., 2016; Cortés

and Margulis, 2017), and it should provide a good estimate of snow on the ground that can be used for hydrological modelling.

Lines 288: "...and is in the order of the model uncertainties (see Figure 2)." You mean 34 mm/yr in comparison to 49.9 mm of RMSE? Please clarify.

Yes. We have now included the number 49.9 in that sentence (section 4.1.2, lines 311-313):

"…, which is similar to the estimates of sublimation amounts for this region (Corripio, 2003; Ayala et al., 2017a, 2017b), and is in the order of the model uncertainties (49.9 mm w.e. in Figure 2a)."

Lines 317-319: "We suspect that this is an expression of the fact that some of the processes not included in TOPKAPI-ETH (namely permafrost, sublimation, snow dynamics or geothermal fluxes) may play a role governing the mass balance of these glaciers". Then it is partially contradictory to this sentence: "...which is a reasonable estimate of sublimation amounts for this region...".

We apologise if we have not been clear in any of those two sentences.

The second sentence refers to the amounts of snow that we remove from the simulations of the Maipo River Basin at the end of each year. We recall that the 1-km resolution simulations of the Maipo River Basin do not consider glaciers. Given the amount and location at high-elevation sites of the removed snow, we think is reasonable to attribute them (at least partly) to sublimation losses. In contrast, the first sentence refers to the models for the individual glaciers, where sublimation is not included.

In relation to neglecting sublimation in the individual models, we include the following sentence in section 4.1.3 (lines 339-343):

"In contrast to the model setup for the entire Maipo River Basin, in this setup we do not perform any corrections to account for sublimation or other mass removal apart from melt. However, as these models are calibrated on volume loss (thus including both losses by sublimation and melting), it can be assumed that glacier response is well captured, but the portioning of hydrological fluxes (sublimation versus runoff) is unconstrained."

Lines 333-335: "Interestingly, several of the glaciers show a positive or near-neutral mass balance over the entire period, which might be an indication that these glaciers have already retreated close to a new equilibrium." This seems to contradict evidence of glacier mass balances in the entire Andes.

We have extended the arguments that might explain the positive or near-neutral mass balance using the following sentence (section 4.1, lines 366-370):

"Interestingly, several of the glaciers show a positive or near-neutral mass balance over the entire period, which might be an indication that these glaciers have already retreated close to a new equilibrium. However, this is not the general trend (as shown by the average values in Figure 4) and it is limited to some specific cases where glaciers have retreated to elevations above the basin-average ELA, or have been covered by thick debris."

Lines 410: Authors report an important and larger ELA elevation than reported in Carrasco et al (2005). It should be highlighted.

We thank the reviewer for this very good comment and pointing out these numbers. We realized that we wrote a different number in the text (370 m and 66 m decade$^{-1}$) than in Figure 6 (239 m and 39 m decade$^{-1}$). The correct

numbers are those in Figure 6, i.e. +239 m (39 m decade$^{-1}$) in the study period. The numbers in the text correspond to an earlier version of our calculations.

In the revised version, we also highlight the differences of our results from those of Carrasco (Section 5.1, lines 449-450):

"These estimates of the ELA change are larger than those calculated by Carrasco et al. (2005), who estimated an increase in the elevation of the 0°C isotherm of about 160 m for central Chile in the period 1975-2001."

Lines 419-410: "In general, glaciers in southern catchments show more positive mass balance than those in northern catchments." This occurs despite elevations are much lower. Any explanation other than precipitation?

This is a very good comment. As shown in Figure 1, the southern catchments (Yeso, Volcán and Upper Maipo) contain a larger proportion of debris-covered and rock glaciers than those in the north (particularly Olivares catchment), which together with precipitation differences can explain the more positive or neutral mass balances.

We have briefly added this (section 5.1, lines 460-461):

"This can be explained by larger precipitation amounts and a higher proportion of both debris-covered and rock glaciers."

Lines 424-440: This is the core of this research. It compares the contributions of ice, snow and rain in annual and summer basis. Is the 3% decrease of glacier summer contribution (entire study period versus current drought) a possible indication peak water was already reached?

Yes, we think that that 3% decrease can be an expression of peak water. However, note that glacier runoff provides 59±23% of the summer runoff in the catchment, and that a 3% decrease is far smaller than the range of the inter-annual variability (23%). We think that a decrease in glacier runoff is more evident in figures 9c and 10.

Lines 441-455 & Figure 9c: Maybe a "realistic" projection could have complemented this analysis.

Yes, we agree that more realistic projections that are forced by global climate simulations should be made for this catchment. However, this study focuses on understanding past changes. Please see also the reply to the main comment (number 1) of reviewer 2.

Lines 481-485: As raised by the authors, difference in mass balances among sub-basins can depend on many climatic and morphological factors, however it is doubtful that precipitation increases that much in semiarid Andes to lead positive mass balances in southern basins. Unless there is data enough to support this statement.

Please note that, according to the CR2 dataset (DGA, 2017), precipitation differences between the northern and southern sub-catchments are in fact very large: up to about 100%, as shown in the next figure.

[Figure]

Annual precipitation from the CR2 product (DGA, 2017). The black frame shows the area where the Maipo River Basin is located.

**TECHNICAL CORRECTIONS**

Line 164: " a meaningful 1955 ice thicknesess..." Delete "a"

We have deleted "a".

Lines 256-261: I think this sentence repeats information from section 3.3.

Not exactly. While in section 3.3 we present the data, in section 4.1.2 we provide more details about how the data are used.

Lines 514-542 Uncertainties of modelling I particularly find this could have been assessed in summary at the end of methods section.

We agree with the reviewer that in its present form, the location of this section was not the best (also noticed by the other reviewer). Our aim is to provide an integrated discussion of the uncertainties of our results within this section. Therefore, we prefer to keep the information at its current location, but we have improved its embedding. We do so by incorporating results from the previous sections, and comments from both reviewers.

Figure 1 (a): Maipo outline may be better recognised if Chile and Argentina are just outlined (without color filling);

We have deleted the colour filling and kept only the outlines of the two countries.

(b) debris-free areas could be coloured in blue because white is difficult to distinguish over yellow;

We noted that the yellow colour of the Volcán sub-catchment corresponded to an old version of that figure. In the revised version, Volcán is coloured in dark orange. Because of this change, we keep the debris-free areas coloured in white. In any case, we improve the visibility of this figure by making some additional changes.

(c) I would recommend sub-basins labels to be horizontally oriented with brackets, so far I cannot tell where are the boundaries between them;

We have oriented the labels horizontally, but we do not clearly see the advantage of using brackets. Please note that there is no clear boundary between the sub-catchments, because the latitude of the glaciers overlap.

(d) Why Volcán label and number of glaciers are in light grey?

The colour should be light blue. We have checked again the colours.

Legend Figure 1: "a) Maipo River Basin next to the city of Santiago, in central Chile; (b) the basin outlet and the sub-catchments, rivers, main glaciers, and hydro-meteorological stations..." These are not all glaciers, nor the 26 modelled glaciers, just main ones.

We have added the term "main glaciers", as suggested by the reviewer.

Please see the new figure 1 below.

[Figure]

Figure 1: a) Maipo River Basin next to the city of Santiago, in central Chile; (b) the basin outlet and the sub-catchments, rivers, main glaciers, and hydro-meteorological stations; (c) the elevation range of every glacier in the basin as a function of the average latitude (arbitrary scale) in each sub-catchment, and the mean elevation (black line); (d) estimated total ice volume using the method developed by Huss and Farinotti (2012) (left axis), and glacierized area (right axis) in each sub-catchment. The surface and glacier type (debris-free, debris-covered or rock glacier), as well as the number of glaciers in each sub-catchment are indicated.

Figure 2 (a): It is Cortes et al 2016 or Cortes and Margulis 2017? Please clarify.

We thank the reviewer for noting this. The correct reference is that of Cortés and Margulis (2017).

Figure 3. I am not sure if this is necessary (instead they could be shown in Figure 1(a). In any case, glaciers in white are difficult to distinguish over yellow. Maybe blue is more appropriate. Name of main glaciers could be added.

As Figure 1a already contains large amounts of information, we would prefer to keep Figure 3 in the article. We have changed the yellow colour of Volcán sub-catchment and use blue for selected glaciers. As suggested by the reviewer, we have added the name of the main glaciers. Please see the new Figure 3 below.

[Figure]

Figure 3: Location of the 26 glaciers modelled with TOPKAPI-ETH. The glaciers' names are given in light blue font. The two black boxes highlight the volcanic areas on which some large glaciers were discarded from the modelled sample.

**Response to reviewer 2**

This article aims to quantify the evolution of the glaciers and the runoff since 1955 in an Andean Chilean catchment using the TOPKAPI-ETH hydro-glaciological model. This study is very interesting and could help for water management in this region. Nevertheless, some issues have to be resolved before publication in the TC journal (see below).

We thank the reviewer for his/her thorough evaluation of our article, for highlighting the relevance of our study, as well as for his/her useful feedback and comments. We have responded to all the questions raised by the reviewer. In the few cases where we did not perform the suggested analyses or simulations, we have provided a detailed justification.

1.  I am not convinced by the long term simulations that use a 'stationary climate' during two decades. By nature, climate is non stationary and an important decadal variability is observed in this region for the different climate variables (for instance the precipitation). The simulations provided here can be a first approach but simulations based on future climate projections should be made. This is important if one consider that this kind of study is oriented to water resource management (as state in various places in the article).

We agree with the reviewer about the importance of climate variability for glaciers and water management, particularly in this region where a large inter-annual climatic variability has been observed. We also agree that projections forced by global climate simulations should be made for this catchment. However, the focus of this study is *on past changes* in glacier and hydrological response, with the aim to understand their drivers. As noted in the text (e.g lines 84, 445, 492, 575, 859 and 867), the committed ice loss scenarios presented in this study are not meant to represent future projections. Rather, they are used for i) understanding how far the glaciers are from an equilibrium after the climatic changes that took place in the period 1955-2016, and ii) providing a baseline for the future changes in hydrology that the basin will experience in any case, i.e. even in the hypothetical case that climate change was to stall. Such committed ice loss scenarios have been increasingly used in the glaciological literature (e.g. Mernild et al., 2013; Christian et al., 2018; Marzeion et al., 2018; Zekollari et al., 2019), but have not been used for hydrological implications so far.

It is for these reasons that we have not performed future simulations. In the revised version, in addition to the limitations already stated in the text, we clearly highlight the aims of our committed ice loss scenarios in the Introduction section (lines 82-87).

 "Additionally, we estimate glacier changes under synthetic scenarios of committed ice loss, in which air temperature, precipitation and cloudiness are assumed to stay at their current levels until the end of the century. We use these scenarios for i) understanding how far the glaciers are from an equilibrium after the climatic changes that took place in the period 1955-2016, and ii) providing a baseline for the future changes in hydrology that the basin will experience in any case, i.e. even in the hypothetical case that climate change was to stall. They are thus highly conservative and do not correspond to a realistic projection for the future."

2.  The methodology to calculate volume and surface glacier variations in relation with the climate is confused. More details should be done (time step, kind of processes, basal sliding, etc....).

The reviewer most likely refers to the methods described in section 4.2 "Extrapolation". In this section, we extrapolate the mass balance of the 26 selected glaciers to all glaciers in the catchment using the methods developed by Huss (2012) for the European Alps. These balances are then used to calculate volume and area variations by means of volume-area scaling.

The method consists of a statistical extrapolation that does not consider any specific physical process (such as basal sliding) explicitly. Such processes are indirectly considered in the uncertainty associated with parameter "c" of the volume-area scaling relation, as explained by Bahr et al. (2015): "...basal sliding and other boundary conditions cannot change the scaling exponent as discussed above, but these boundary conditions could have a very important influence on the random distribution of c.".

To clarify these issues, we include these changes in the revised version (Section 4.2):

- We add the word "annual" in the method description to clarify that these calculations have an annual time step.
- In relation to the processes mentioned by the reviewer, we include this sentence (lines 398-400): "The uncertainty in parameter c should indirectly account for the different boundary conditions (such as basal sliding or surface geometry) that are found at each glacier (Bahr et al., 2015)."

3. Concerning the precipitation used in the model, a clear explanation on how the discrimination phase between solid and liquid is done is missing.

This information was included in lines 225-226 but might have not been clear enough. In the revised version, we write (section 4.1.1, lines 241-243):

"The model simulates snowfall at a given grid cell when precipitation occurs and air temperature is below a threshold parameter. If air temperature is above that threshold, precipitation is considered as rain."

The calibrated values of the precipitation threshold parameter are given in Table 1. The values are 0°C for the individual glaciers and 2°C for the Maipo River Basin to account for the different spatial resolutions and extents of the different models.

I don't understand why an additional meteorological station is used here. Only one station is not adequate to the size of the catchment.

There seems to be a misunderstanding here. As explained above, we do not use a single station to discriminate between snowfall and rain but rather use the temperature at each grid cell at the time of the precipitation event. If the reviewer refers to the extrapolation of air temperature from the Embalse El Yeso station, we have added the following sentence (section 6.3, lines 574-577):

"An additional simplification in the meteorological distribution is the extrapolation of air temperature from one single station. Nevertheless, we are confident that air temperature variability is well constrained over the catchment, because it usually correlates well over long distances, daily lapse rates are derived from the basin-wide CR2 temperature dataset, and the timing of snow disappearance is well simulated by TOPKAPI-ETH."

A correction is made on the raw precipitation but details should be given concerning the methodology used.

We have added the following details in relation to the precipitation correction (section 4.1.2, lines 297-299):

"We obtain a precipitation correction factor by manually fitting the observed and modelled curves of SCA and SWE, and at the same time closing the water balance of the basin. We obtain a value of +50%."

To improve the justification of this relatively large value, in addition to the errors originated from the reanalysis data (already discussed in the original manuscript), we include undercatch as a possible explanation (lines 295-297):

"Although the CR2 precipitation product corrects the ERA-Interim values by comparing them with ground data, these data are available only below 3000 m a.s.l. in this region, and have not been corrected for gauge undercatch (DGA, 2017), which can also contribute to the underestimation of precipitation at the highest elevations (Rasmussen et al., 2012)."

Finally how does the model compute the sublimation?

This is a very good question. We think that our explanations might have not been clear enough. Surface sublimation is not computed in the TOPKAPI-ETH model. To avoid confusion, this is now clearly stated in the model description of the revised manuscript (Section 4.1.1, line 254):

"TOPKAPI-ETH does not compute sublimation."

Additionally, we have extended the discussion of the issues caused by neglecting sublimation in the simulations for the entire Maipo River Basin (section 4.1.2, lines 306-315):

"An additional aspect of model simplifications identified during the model calibration is that air temperature over areas above 5000 m a.s.l. (about 5% of the basin) is most of the time lower than the air temperature threshold parameter for melt onset, generating large snow accumulation that is not seen in the SWE reconstruction product. As snow on this high-elevation areas is in reality removed by wind transport and sublimation, we reset the SWE in the model to zero at the beginning of each hydrological year. Although this implies that the model is not strictly mass-conserving, we verify that the discarded snow is in average 34 mm yr$^{-1}$ over the entire basin (or 688 mm yr$^{-1}$ = 1.9 mm d$^{-1}$ over the areas above 5000 m a.s.l.), which is a reasonable estimate of sublimation amounts for this region (Corripio, 2003; Ayala et al., 2017a, 2017b), and is in the order of the model uncertainties (49.9 mm w.e. in Figure 2a). As elevation decreases south, the discarded snow varies from about 121 mm w.e. yr$^{-1}$ over the Colorado sub-catchment to about 10 mm w.e. yr$^{-1}$ over Upper Maipo."

In the setup for the individual glaciers, we add the following comment (section 4.1.3, lines 339-343):

"In contrast to the model setup for the entire Maipo River Basin, in this setup we do not perform any corrections to account for sublimation or other mass removal apart from melt. However, as these models are calibrated on volume loss (thus including both losses by sublimation and melting), it can be assumed that glacier response is well captured, but the portioning of hydrological fluxes (sublimation versus runoff) is unconstrained."

4. For the snow cover evolution, no in-situ data is provided. Does such data exists? If yes, a comparison between the simulations of the snow cover with TOPKAPI and CAMEL-CL models should be made. Please give more details concerning the CAMEL-CL product (resolution, etc...).

Our simulation of SWE in the catchment has been calibrated and validated using the SWE reconstruction of Cortés and Margulis (2017), included in the CAMELS-CL database. In the revised version, we provide more details about these products.

Section 3.4 (lines 217-225):

"These basin-scale SWE estimates were aggregated by Álvarez-Garretón et al. (2018) from a daily gridded SWE reconstruction for the Andes Cordillera generated by Cortés and Margulis (2017) at a 180-m resolution. The SWE reconstruction was obtained from a data assimilation framework that integrates a land surface and depletion model, the assimilation of Landsat imagery, and the Modern Era Retrospective Analysis for Research and Applications (MERRA) reanalysis as a forcing dataset (Cortés et al., 2016; Cortés and Margulis, 2017). Although not all physical processes are included in the assimilation process (for example, blowing snow sublimation), the dataset has been validated at several sites across the Southern Andes (Cortés et al., 2016; Cortés and Margulis, 2017), and it should provide a good estimate of snow on the ground that can be used for hydrological modelling."

To further extend the validation of our model results, we add a new sub-section in the supplementary material ("Section S3: Additional model validation"). There, we include the comparison of direct measurements and simulated values of SWE at the DGA (Chilean Water Directory of Water Resources) monitoring site of Laguna Negra. Due to its length (5 pages), the new sub-section is given at the end of this document.

5.  Recent studies underlined the importance of groundwater in mountainous catchments. Here in the model, it seems that no water flux into the ground exists. This cannot be true. Subterranean water fluxes may have an importance for the future of water resources.

We are aware that several studies have been uncovering the role of groundwater, both in sedimentary and in fractured rock systems, in mountainous catchments. TOPKAPI-ETH does indeed simulate sub-surface water fluxes (possibly our description at lines 245-248 of the original manuscript was too brief as to be noted), albeit in a simpler way than many dedicated groundwater models. Since we focus mainly on the snow and ice mass balance components of the water cycle, and given that we are able to validate the simulation of these components independently, we believe that the uncertainty associated with groundwater fluxes should not affect significantly the conclusions of our work. Additionally, the comparisons with observed streamflow in our work are conducted at stations located in narrow gorges with rock outcroppings, where subsurface fluxes should be minimum compared with surface river flow.

6.  Please define 'glacier runoff'.

Glacier runoff is defined as the sum of rain, snowmelt and ice melt generated in the areas defined by the glaciers outlines in 1955. We acknowledge that this was stated relatively late in our article (lines 367-368). In the revised version, we provide this definition in the Introduction (Section 1, line 81).

7.  If the model is oriented to be used for water management (as stated in the lines 544-547), please give results for daily simulations. What is the agreement between the simulations and the observations at daily time-step?

In the revised version, we provide an evaluation of simulated streamflow at a daily time step. However, since daily streamflow records at the outlet are not corrected for water extractions or the operation of the Embalse El Yeso dam (in opposite to the monthly records we used for calibration), we do not provide a direct comparison. Instead, we provide a comparison based on flow-duration curves. As suggested by reviewer 1, we also provide a comparison of model results with streamflow records at intermediate gauges.

All this information is included in the new sub-section of "Additional model validation" in the supplementary information of the article (also included at the end of this document).

8.  I think that all the sections 6.3, should be moved at the beginning of the result section.

Reviewer 1 raised a similar comment. Section 6.3 aims at providing an integrated discussion of the uncertainties in our results, and we therefore prefer to improve the text rather than reshuffle its location. We do so by incorporating results from the previous sections, and comments from both reviewers.

**Specific comments:**

Abstract – line 14: please precise the time step of the simulated runoff

We have added this information (line 18):

"TOPKAPI-ETH is run at a daily time step using…"

Abstract – line 20: please precise the latitude range

We have added this geographical information (lines 14-15):

"We investigate glacier runoff in the period 1955-2016 in the Maipo River Basin (4 843 km2, 69.8-70.5°W, 33.0-34.3°S), semiarid Andes of Chile."

Abstract – Please precise if the glacier area's changes are taking into account in the model

TOPKAPI-ETH does take into account glacier area changes. To make this more explicit, we have added this sentence (lines 16-17):

"We model the mass balance, area and volume changes, and runoff contribution of 26 glaciers with the physically-oriented and fully-distributed TOPKAPI-ETH glacio-hydrological model, and extrapolate the results to the entire basin."

This is also better explained in the model description of the revised version (section 4.1.1, lines 264-265):

"Negative annual mass balances can result in glacier area reductions, but no area increases due to positive mass balances are prescribed. Area changes are applied at the end of March."

Line 81: "....estimate glacier changes...." please precise if it is surface, volume or both ?

In this line of the revise version we write (lines 79-81):

"Our main objective is to reconstruct glacier changes (area and volume) during the last six decades in one of the main catchments of the semiarid Andes, the Maipo River Basin, analyse the role of glaciers in the regional hydrology, and identify the main trends in glacier runoff."

Line 95: "...., to which it provides most of the drinking water…" please specify the percentage (give a quantity).

We have now specified the percentage (lines 96-97):

"The basin is located in central Chile (~33°S, ~70°W), to the east of the Chilean capital city, Santiago (Figure 1a), to which it provides about 70% of its drinking water (DGA, 2004)."

Line 125: If I understand well, the outlines taken in 1955 are used for the year 2000 ? If the answer is yes, it is certainly not true. Please add more details.

We did not use the 1955 outlines for the year 2000. The year 2000 is the year the SRTM DEM refers to. We intersected that DEM with the outlines of the national glacier inventory, which was derived using images from 2003.

We have now explained this as (lines 127-128):

"For consistency with the DEM obtained from the Shuttle Radar Topography Mission (SRTM), we assume that the outlines in the national inventory from 2003 are also valid for 2000"

Lines 191-194: "....for the years 2004-2016....." How do you do before 2004?

This procedure is explained in the next paragraph of the article (lines 206-209):

"Values for air temperature gradients and cloud transmissivity in the study periods without information from CR2 and the Chilean solar radiation database (1955 to 1978 and 1955 to 2003, respectively) are randomly selected from a pool of values recorded in the same day of the year in the periods with available information."

Line 221: The model is "physically-oriented", so how do you do with the land cover and land use changes over the last decades? Please indicate clearly that in the article. If the changes are important, this should be taken into account in the model.

We thank the reviewer for this very good comment. We agree with the reviewer that land cover and land use changes can impact the hydrological simulations.

In the original article we explain how land use was derived in section 3.4 (lines 213-214):

"For modelling evapotranspiration and sub-surface water fluxes, we generate land use and soil types maps, respectively. The land use maps are extracted from the National Forest Corporation (CONAF) database (CONAF, 2013),…"

Unfortunately, to our knowledge, there are no data available to evaluate meaningful changes in land cover. We have now included that information (section 3.4, 230-232):

"To our knowledge, there are no enough detailed datasets to evaluate changes in land use throughout the study period, and we keep land use and soil types constant in our simulations."

Line 245: "....but no area increases due to positive mass balance are prescribed." This is not a valid statement as it is possible to observe glacier's advances. So if the model is "physically-oriented" this should be changed.

TOPKAPI-ETH has been defined as "physically-oriented" because it represents the main glacio-hydrological processes with the most relevant variables of each process. Examples are the ETI model for snow and ice melt (Pellicciotti et al., 2005), and the SnowSlide model for gravitational distribution of snow (Bernhardt and Schulz, 2010). Glacier advances are particularly difficult to model because they require an explicit simulation of ice flow, requiring information on ice rheology, basal sliding and internal deformation. The explicit simulation of ice flow would increase the computational cost to an extent that is not compatible with the purposes of this particular modelling exercise. Alternatively, ice flow models applicable to the basin scale have emerged only very recently (e.g. Zekollari et al., 2019), and are not included in TOPKAPI-ETH for the time being.

However, the deviations associated with positive glacier area and volume changes are implicitly taken into account by assuming an uncertainty in the volume-area scaling parameter "c". In any case, apart from some glacier advances in the 1980s and 2000s (Masiokas et al., 2016), no generalized or long-term advance phase has been documented in this region, so neglecting this process has a limited effect. Please see also our reply to comment 2.

In all this part, the time-step should be precised.

In the revised version we include:

- The time steps at which the model can be run when describing the model (Section 4.1.1, line 239)
    "…can be run at different spatial and time steps (typically hourly or daily),"
- The time step we used (daily) in our setups
    o (Section 4.1.2, lines 274-275): "The model is run continuously from 1955 to 2016 at a daily time step."
    o (Section 4.1.3, line 325): "The models are run at a daily time step starting in the year 1955,…"

In the model, the selected calibration and validation periods are unclear.

We have added these periods (section 4.1.2):

"The calibration of the Maipo River Basin model was performed for the period April 2003 to March 2016, and…"

"…, we use the period April 1984 to March 2003 for model validation."

Furthermore, details should be given concerning the snowfall/rainfall discrimination (T°threshold ?).

Please see our reply to comment 3.

I don't understand why the ERA-interim and MERRA products are not tested in the model. Please explain why.

In our study, we use the CR2 products of precipitation and air temperature, which were computed using a statistical downscaling of ERA-Interim variables (lines 182-188, given here below). On the other hand, the SWE product is a reanalysis obtained through a combination of an energy balance model forced with MERRA, with data assimilation of Landsat snow cover. We operate under the assumption that the downscaled products are a better representation of the local conditions than ERA-interim and MERRA can be, as most of the local meteorological information have been used for downscaling CR2. A further validation of ERA-Interim and MERRA is beyond the scope of this study.

This information is present in the manuscript (Section 3.3, lines 190-196):

"The CR2 daily precipitation product was generated by means of a statistical downscaling of precipitation and moisture fluxes from the ERA-Interim reanalysis. The downscaling procedure is based on multiple linear regressions with topographic parameters, which were calibrated with quality-controlled precipitation records. The CR2 temperature product was obtained using near-surface temperature from ERA-Interim and land surface temperature (LST) from the Moderate Resolution Imaging Spectroradiometer (MODIS), by means of multiple regression models using LST as the explanatory variable and validated with local observations."

Line 315: What is the criterion "to fit the geodetic mass balances…"?

We have now specified the criterion as (section 4.1.3, lines 335-339):

"Glacier-wide mass balance is considered as fitted when the difference between the simulated and observed balance is smaller than a certain threshold. We find that choosing a threshold equal to half of the uncertainty in the geodetic mass balance allows for reliable simulations while keeping an acceptable computation time. The uncertainty of the geodetic mass balances is 3.2 and 1.2 m w.e. for the periods 1955-2000 and 2000-2003, respectively. "

Line 366 – 367:  Please rewrite this sentence, unclear.

We change the original sentence: "The uncertainty in glacier runoff is estimated at each year as proportional to that calculated for glacier volume." into "At a particular year, the uncertainty in glacier runoff is estimated as a fraction of the same variable. That fraction is the same as that between glacier volume and its uncertainty in that year."

Fig. 1: please specify in the legend how the total ice volume is obtained

We include this information in the new caption of Figure 1:

"… (d) estimated total ice volume using the method developed by Huss and Farinotti (2012)"

Fig.6: you should add the uncertainty for each curve.

We have added an uncertainty band for the extrapolation of mass balance to the entire basin (Figure 6c). We did not do it for the external data (precipitation and temperature from CR2 products, and Echaurren Norte mass balance from DGA), because uncertainty is not provided in the datasets. The uncertainty in the mass balance of the sub-catchments is shown in the new Table 4 (suggested by reviewer 1).  Please see the new Figure 6 below.

[Figure]

Figure 6: Variability of meteorological and glaciological variables in the Maipo River Basin over the period 1955-2016. (a) Air temperature and precipitation with a 3-year moving mean, (b) equilibrium line altitude (ELA), (c) cumulative glacier mass balance for the modelled glaciers (simulated with TOPKAPI-ETH), the entire basin (extrapolation) and its associated uncertainty, and the measurements on Echaurren Norte Glacier, and (d) cumulative glacier mass balance for each sub-catchment. In b), the difference between the ELA in the last 10 years (2006-2016) and the first 10 years (1955-1965) of the study periods is indicated, as well as the equivalent ELA increase rate. The shadowed area in (b) shows the standard deviation of the elevation of grid cells with a mass balance between –0.1 m w.e and 0.1 m w.e.

Fig. 7: Where is the subterranean part ?

In TOPKAPI-ETH, subsurface and groundwater flux components are routed and added to the total flow at every sub-catchment closing point. As such, they are considered in the total flow volumes and compared against river streamflow observations. In general, as these gauging stations are placed at locations where much of the overall

basin flow is captured, there is not a major subterranean component to flow. Please see also our response to comment 5.

Please indicate the evolution of glacier volume and glacier area.

This information is found in Figure 9 and we don't think that repeating it here would be beneficial.

Fig. 8: In the figure 7, you indicate Rain = 3% and in the figure 8 you indicate Rain : 29+/-8 % , why?

While Figure 7 shows the runoff contribution from the area that was glacierized in 1955, Figure 8 shows the runoff contribution in the entire Maipo River Basin.

Tab. 1: Please indicate the tested ranges for each parameter and the references associated.

We have now provided the ranges and references. Note that we used ranges for only some of the parameters. For the rest of the rest of the parameters we used typical values from the literature since they showed good performance.

While our simulations of SWE at Laguna Negra compare well to the observations (Fig. S6-S7), this is not always the case for the streamflow values. This is partly because there are many water diversions that subtract water from the Maipo River and its tributaries, and some of the available streamflow records at intermediate gauges have not been corrected for these water extractions. However, considering that no specific calibration of the sub-surface parameters for the intermediate river sections was performed, results of the model validation are in general satisfactory at both monthly (Tab. S3 and Fig. S3-S4) and daily (Fig. S5) time scales.

**Table S3: Results of the model validation at streamflow gauges at the monthly scale**

| Gauge | Time period | Average streamflow ($m^3 s^{-1}$) | Nash-Sutcliffe (NS) | Root Mean Square Error (RMSE) ($m^3 s^{-1}$) | Mean Bias (BIAS) (%) |
|---|---|---|---|---|---|
| Río Maipo en Las Hualtatas | 1979-2013 | 32.3 | 0.63 | 15.6 | -11.2 |
| Río Volcán en Queltehues | 1955-2015 | 8.1 | 0.61 | 6.1 | -22.0 |
| Río Maipo en San Alfonso | 1955-2015 | 73.0 | 0.60 | 35.8 | -12.2 |
| Río Colorado antes junta río Olivares (*) | 1978-2016 | 11.3 | 0.49 | 9.9 | 23.9 |
| Río Olivares antes junta río Colorado (*) | 1978-2016 | 6.0 | 0.26 | 7.0 | -4.8 |
| Río Colorado antes junta río Maipo (*) | 1955-2015 | 30.3 | 0.26 | 17.4 | 19.6 |

(*): Available streamflow observations are not corrected for water extractions

[Figure]

| Number | Name |
|--------|------|
| 1 | Río Maipo en Las Hualtatas |
| 2 | Río Volcán en Queltehues |
| 3 | Río Maipo en San Alfonso |
| 4 | Río Colorado antes junta río Olivares |
| 5 | Río Olivares antes junta río Colorado |
| 6 | Río Colorado antes junta río Maipo |
| 7 | Laguna Negra |

Figure S2: Location of intermediate streamflow gauges and the Laguna Negra snow monitoring site

[Figure]

Figure S3: Validation of model results at six intermediate streamflow gauges. Monthly time series.

[Figure]

Figure S4: Validation of model results at six intermediate streamflow gauges. Flow-duration curves of monthly time series.

[Figure]

Figure S5: Validation of model results at six intermediate streamflow gauges. Flow-duration curves of daily time series.

[Figure]

Figure S6: Validation of model results using SWE manual measurements at Laguna Negra (33.67°S, 70.11°W). The SWE measurements consist of 50 data points measured in the period 1969-2007. Daily time series of simulated values against observations.

[Figure]

Figure S7: Validation of model results using SWE manual measurements at Laguna Negra (33.67°S, 70.11°W). The SWE measurements consist of 50 data points measured in the period 1969-2007. Scatter plot of observations and simulated values at the time of the measurements.

[revised manuscript text omitted]

---

## Editor Decision (ED1)

Thank you for your thorough response to the reviewer's comments and thorough revision of your manuscript. The paper was reviewed again by one of the reviewers and they were impressed with your improvements. There are a small number of technical issues (see below) which should be corrected.

I look forward to receiving the final version of the manuscript.

Best regards,

Dr Liz Bagshaw

Please be aware of the discrepancy of the numbers between your response letter and the manuscript regarding the following sentence, just in case any possible misspelling had been transferred into the manuscript.
- In response letter:
"As elevation decreases south, we estimate that sublimation varies from about 20 mm w.e. yr-1 in the Colorado sub-catchment to about 3 mm w.e. yr-1 in the southern ones (using values normalized by the entire Maipo River Basin area)."
- In the manuscript:
"As elevation decreases south, the discarded snow varies from about 121 mm w.e. yr-1 over the Colorado sub-catchment to about 10 mm w.e. yr-1 over Upper Maipo."

The reviewer raises the following minor technical corrections:
- Figure 1: keeps saying glaciers.
- Section S1: That stereoscopically reconstructs
- Figure S2: There is an obvious overlap of st. Rio Olivares antes junta Colorado & Rio Colorado antes junta Olivares, just raise that issue on legend.

Figure 2c: the legends overlap the plots such they are hard to read. Please adjust the spacing.

---

## Author Response (AR2)

**Glacier runoff variations since 1955 in the Maipo River Basin, semiarid Andes of central Chile**

Response to Editor

Editor: Black font

Authors: Blue font

Thank you for your thorough response to the reviewer's comments and thorough revision of your manuscript. The paper was reviewed again by one of the reviewers and they were impressed with your improvements. There are a small number of technical issues (see below) which should be corrected.

I look forward to receiving the final version of the manuscript.

Best regards,

Dr Liz Bagshaw

We thank the Editor very much for her comments and guidance during the review process. We have corrected the technical issues pointed out by the reviewer, updated a few references, and submitted the final version of the manuscript. Please see our responses to the technical issues below.

Please be aware of the discrepancy of the numbers between your response letter and the manuscript regarding the following sentence, just in case any possible misspelling had been transferred into the manuscript.

- In response letter: "As elevation decreases south, we estimate that sublimation varies from about 20 mm w.e. yr-1 in the Colorado sub-catchment to about 3 mm w.e. yr-1 in the southern ones (using values normalized by the entire Maipo River Basin area).".

- In the manuscript: "As elevation decreases south, the discarded snow varies from about 121 mm w.e. yr-1 over the Colorado sub-catchment to about 10 mm w.e. yr-1 over Upper Maipo."

We thank the Editor for noticing this issue. The correct numbers and text are those in the manuscript, whereas those in the response letter were extracted from a previous version of the calculation. The second text is also more precise about the origin of those numbers.

The reviewer raises the following minor technical corrections:

We thank the reviewer very much for the new evaluation of our article. Please see our detailed responses below.

- Figure 1: keeps saying glaciers.

Corrected. We changed "glaciers" to "main glaciers" as suggested in the first round of reviews.

- Section S1: That stereoscopically reconstructs

Corrected. We changed "That stereoscopically reconstruct" to "That stereoscopically reconstructs".

- Figure S2: There is an obvious overlap of st. Rio Olivares antes junta Colorado & Rio Colorado antes junta Olivares, just raise that issue on legend.

We added "Note that gauges 5 and 6 are closely located." to the figure caption.

Figure 2c: the legends overlap the plots such they are hard to read. Please adjust the spacing.

Corrected. We adjusted the spacing to avoid the overlap between the legend and lines.

Finally, we also updated the reference to the article in which the geodetic mass balances used in our study are presented. The new reference is:

Farías-Barahona, D., Ayala, Á., Bravo, C., Vivero, S., Seehaus, T., Vijay, S., Schaefer, M., Buglio, F., Casassa, G. and Braun, M. H.: 60 years of glacier elevation and mass changes in the Maipo River Basin, central Andes of Chile, Remote Sens., In review, 2020.